# ATLASKV: AUGMENTING LLMS WITH BILLION-SCALE KNOWLEDGE GRAPHS IN 20GB VRAM

**Haoyu Huang[1], Hong Ting Tsang[1], Jiaxin Bai[1,\*], Xi Peng[2], Gong Zhang[2], Yangqiu Song[1]**

[1]The Hong Kong University of Science and Technology
[2]Theory Lab, Huawei

{hhuangcp, httsangaj, jbai}@connect.ust.hk
{pancy.pengxi, nicholas.zhang}@huawei.com
yqsong@cse.ust.hk

 Source Code     Data and Models

## ABSTRACT

Retrieval-augmented generation (RAG) has shown some success in augmenting large language models (LLMs) with external knowledge. However, as a non-parametric knowledge integration paradigm for LLMs, RAG methods heavily rely on external retrieval modules and the retrieved textual context prior. Especially for very large scale knowledge augmentation, they would introduce substantial inference latency due to expensive searches and much longer relevant context. In this paper, we propose a parametric knowledge integration method, called **AtlasKV**, a scalable, effective, and general way to augment LLMs with billion-scale knowledge graphs (KGs) (e.g. 1B triples) using very little GPU memory cost (e.g. less than 20GB VRAM). In AtlasKV, we introduce KG2KV and HiKVP to integrate KG triples into LLMs at scale with sub-linear time and memory complexity. It maintains strong knowledge grounding and generalization performance using the LLMs' inherent attention mechanism, and requires no external retrievers, long context priors, or retraining when adapting to new knowledge.

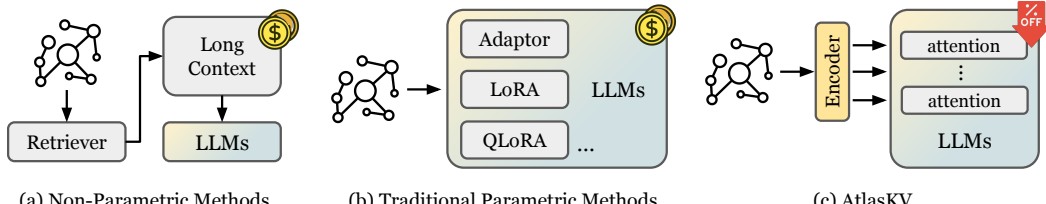

(a) Non-Parametric Methods        (b) Traditional Parametric Methods        (c) AtlasKV

Figure 1: The simple illustrations of two kinds of popular knowledge augmentation paradigms for LLMs and a new parametric knowledge augmentation paradigm adopted by AtlasKV: (a) Non-parametric methods usually rely on external retrievers and long context prior, which have retriever-limited performance and substantial inference latency. (b) Traditional parametric methods require re-training the model when adapting to new knowledge, which are also expensive. (c) AtlasKV can achieve injecting external knowledge efficiently at scale without need for external retrievers or long context prior with strong generalization ability.

## 1 INTRODUCTION

Large language models (LLMs) have demonstrated impressive generation abilities in various down-stream tasks (Brown et al., 2020; Touvron et al., 2023; Kung et al., 2023; Li et al., 2024; Dagdelen

---

*Corresponding author.

et al., 2024), where their expanding parameter scales enable them to function as comprehensive knowledge stores by encoding factual information directly into their parameters (Petroni et al., 2019; Jiang et al., 2020; Rae et al., 2021; Bubeck et al., 2023; Morris et al., 2025). Retrieval-augmented generation (RAG) (Gao et al., 2023; Fan et al., 2024) is a more cost-efficient solution to enhance the capabilities of LLMs in knowledge-intensive tasks, which may require vast amount of external knowledge, without altering an LLM's parametric representation. RAG methods usually retrieve text chunks (Sarthi et al., 2024; Jimenez Gutierrez et al., 2024) or subgraphs (Edge et al., 2024; Huang et al., 2025) that are relevant to a query from an external textual knowledge base (KB) or knowledge graph (KG), which serve as the context prior for LLMs to generate responses.

Although RAG methods have achieved some success in efficiently augmenting LLMs with external knowledge, they still face some critical limitations. As shown in (a) of Figure 1, these non-parametric methods heavily rely on external retrieval modules and the textual context prior, which introduce substantial inference latency due to expensive searches (e.g. nearest-neighbor searches) (Khandelwal et al., 2019; He et al., 2021a; Shi et al., 2023) and longer context (Ram et al., 2023; Cao et al., 2025) specially for very large scale knowledge augmentation.

In contrast, parametric approaches (Gururangan et al., 2020; Wang et al., 2024; Cao et al., 2025) can achieve the integration of external knowledge into LLMs without need for external retrievers or long context prior. Traditional knowledge adaptation techniques (Hu et al., 2022; Diao et al., 2023), as shown in (b) of Figure 1, require retraining the model when adapting to a new distribution of knowledge, which significantly limits their application scenarios. A new parametric knowledge augmentation paradigm introduced by KBLaM (Wang et al., 2024) like (c) of Figure 1, well addresses this issue by encoding external knowledge into a series of key-value parametric representations and seamlessly injecting them into the self-attention layers of an LLM, which well preserves the advantages of parametric methods and can also adapt to any new KB in a training-free manner.

Nevertheless, we found that there are two critial challenges of this novel knowledge augmentation paradigm that restrict its real-world applications: (1) **Lack of high quality training data**. It requires query-key-value (Q-K-V) sentences of the external knowledge as the training data. Directly synthesizing Q-K-V training data from unformatted documents with fixed pre-defined schemas suffers from limited query diversity, which could result to poor generalization performance in out-of-distribution (OOD) scenarios. (2) **Poor scalability**. When augmenting LLMs with very large scale external KB or KGs, the computational and memory overhead of this knowledge augmentation paradigm becomes prohibitively high, even with linear time and memory complexity.

To this end, we propose **AtlasKV**, a scalable method that enables end-to-end knowledge augmentation of LLMs with billion-scale KGs (e.g. 1B triples) using very little GPU memory cost (e.g. less than 20GB VRAM) while achieving superior knowledge grounding and generalization performance in OOD scenarios. We achieve this by two innovative designs from both data and algorithm perspectives. Specifically, to address the training data quality issue, we observe that each triple in KGs can be naturally converted into Q-K-V data, which shares very similar structure with the Q-K-V vectors of self-attention networks in LLMs. So we introduce the concept of **KGKV** and propose the **KG2KV** pipeline that naturally converts each KG triple into high-quality Q-K-V data for both training and inference, enabling a better injection of KGs into LLMs. To solve the scalability challenge, we propose a hierarchical key-value pruning (**HiKVP**) algorithm that can dramatically reduce computational and memory overhead while maintaining high knowledge grounding accuracy during inference time.

In summary, we make the following main contributions:

- We propose **AtlasKV**, a scalable method that enables end-to-end augmentation of LLMs with billion-scale KGs (e.g. 1B triples) using very little GPU memory (e.g. less than 20GB VRAM) while achieving superior knowledge grounding performance and strong generalization abilities.

- We introduce **KG2KV** and **HiKVP** as complementary designs to address data and algorithmic challenges respectively: KG2KV naturally transforms KG triples into high-quality Q-K-V data to enhance generalization, while HiKVP enables scalable integration through hierarchical pruning that dramatically reduces computational and memory overhead during inference.

- Extensive experiments and analysis demonstrate the superior effectiveness and scalability of AtlasKV compared to ICL, KBLaM, and RAG methods, with comprehensive ablation studies validating the contribution of each component.

## 2 RELATED WORK

***Non-parametric Knowledge Augmentation Methods For LLMs.*** The most popular non-parametric knowledge augmentation methods for LLMs is RAG (Lewis et al., 2020; Gao et al., 2023; Fan et al., 2024), which significantly enhances LLMs by incorporating an external retriever that fetches relevant context from external KBs (Zhang et al., 2024a; Sarthi et al., 2024) or KGs (Mavromatis & Karypis, 2024; Edge et al., 2024; Jimenez Gutierrez et al., 2024; Huang et al., 2025). Their retrievers usually heavily rely on either the separately pretrained sentence transformers (Wang et al., 2020a; Izacard et al., 2021; Zhang et al., 2025a), or a well-designed tuning process with the LLM's output as the feedback signal (Shi et al., 2023; Chang et al., 2025). However, the performance of this knowledge augmentation paradigm could be significantly limited by the capabilities of the retrievers. There are also some graph-based RAG advancements can scale to large knowledge bases more efficiently. $E^2$GraphRAG (Zhao et al., 2025) constructs bidirectional indexes between entities and chunks to enable fast lookup during both local and global retrieval. LinearRAG (Zhuang et al., 2025) constructs a relation-free hierarchical graph to let graph construction scale linearly with corpus size. Nevertheless, they still follow the ICL-based RAG paradigm. And the long context retrieved from large KBs or KGs could still introduce substantial inference latency.

***Parametric Knowledge Augmentation Methods For LLMs.*** Parametric knowledge augmentation approaches are much more native to LLMs. Because the inherent memory of LLMs is integrated by pretraining and supervised fine-tuning (Geva et al., 2020; Petroni et al., 2019; Morris et al., 2025), which are also parametic methods. Some early attempts to efficiently integrate new external knowledge into LLMs such as LoRA (Hu et al., 2022) and adapter (He et al., 2021b; Diao et al., 2023) still suffer from retraining the model when integrating new external knowledge into LLMs. Cache-augmented generation (CAG) (Chan et al., 2025) is a new caching-based knowledge augmentation paradigm that preloads the LLM with all relevant documents in advance and precomputing the KV cache. MemDec (Cao et al., 2025) provides a domain-specific memory module that enhances various frozen LLMs without parameter modifications. KBLaM (Wang et al., 2024) introduces a new knowledge augmentation paradigm that augments knowledge into the attention layers of LLMs, achieving linear computational complexity while enabling training-free adaptation to new KBs after initial training.

***Knowledge Graph Augmentation Methods For LLMs.*** There are many works augmenting KGs into LLMs in both parametric and non-parametric ways. Except for the graph-based RAG methods mentioned above, some LLM-based KGQA methods like RAR (Shen et al., 2025), KnowGPT (Zhang et al., 2024b), and the work done by Ji et al. (2024) also depend on the training of their retrievers or path aligner to find the most relevant knowledge from KGs as context. KELP (Liu et al., 2024) explores latent semantic matching to improve path-level knowledge selection from KGs. They are still limited by the performance of the retrievers and also face inference latency issues.

## 3 BACKGROUNDS AND DEFINITIONS

### 3.1 KNOWLEDGE GRAPHS

As most of the existing works on graph-based RAG systems did, we use the textual triples $(h, r, t)$ as the basic knowledge unit of KGs in our work, which could be extracted from unstructured text with any existing KG extraction method (Angeli et al., 2015; Huang et al., 2024; Mo et al., 2025). Then the KGs can be defined as $\mathcal{G} = \{(h, r, t) | h, t \in \mathcal{E}, r \in \mathcal{R}\}$, where $\mathcal{E}$ is the set of entities and $\mathcal{R}$ is the set of relations. Note that $h, t$ could be either named entities, or other types of entities, such as concepts, events, etc (Zhang et al., 2020; Tan et al., 2024; Bai et al., 2025). In our work, $\mathcal{G}$ will be integrated as external factual knowledge for LLMs to ground facts and answer questions. And the process of extracting KGs from documents is not the focus of our work.

## 3.2 ATTENTION NETWORKS

In this section, we will first give the definitions of self-attention layers, which are the key components of the transformer (Vaswani et al., 2017) backbone. Then we will describe the definitions of the rectangular attention in KBLaM (Wang et al., 2024), which is a new knowledge augmentation paradigm adopted by AtlasKV.

***Self-Attention Layers.*** In each attention layer, we input a query $x \in \mathbb{R}^{N \times 1}$ with $N$ token length, which embedding vector can be denoted as $\boldsymbol{x}^{(l)} \in \mathbb{R}^{N \times D}$. $D$ is the embedding dimension of attention layers and $l \in \{1, .., L\}$, where $L$ is the number of attention layers. There are also three attention heads $\mathbf{W}_Q^{(l)}, \mathbf{W}_K^{(l)}, \mathbf{W}_V^{(l)} \in \mathbb{R}^{D \times D}$, which are designed to project each input token into Q-K-V embeddings $\mathbf{q}^{(l)}, \mathbf{k}^{(l)}, \mathbf{v}^{(l)} \in \mathbb{R}^{N \times D}$. Then the output at the $l$-th layer and $n$-th token is computed as

$$\mathbf{y}_n^{(l)} = \frac{\sum_{i=1}^n \exp(\langle \mathbf{q}_n^{(l)}, \mathbf{k}_i^{(l)} \rangle / \sqrt{D}) \mathbf{v}_i^{(l)}}{\sum_{i=1}^n \exp(\langle \mathbf{q}_n^{(l)}, \mathbf{k}_i^{(l)} \rangle / \sqrt{D})}, \tag{1}$$

where $\langle \cdot, \cdot \rangle$ denotes the inner product of two vectors. This standard implementation of self-attention can have a time complexity of $\mathcal{O}(N^2 \cdot D)$ and memory complexity of $\mathcal{O}(N \cdot (N + D))$, which could lead to significant computational overheads and time delay as the input length $N$ gets larger.

***Rectangular Attention in KBLaM.*** The KB in KBLaM is a set of key-value pairs, which is denoted as $\mathcal{M} = \{(\boldsymbol{k}^{(l)m}, \boldsymbol{v}^{(l)m})\}_{m=1}^M$ at the $l$-th attention layer, where $\boldsymbol{k}^{(l)m}, \boldsymbol{v}^{(l)m} \in \mathbb{R}^{M \times D_E}$ are the base embedding vectors of the $m$-th key-value pair. $M$ is the size of the KB and $D_E$ is the output dimension of the sentence encoder. Then the knowledge augmented output at the $l$-th attention layer and $n$-th token is computed as

$$\tilde{\mathbf{y}}_n^{(l)} = \frac{\sum_{m=1}^M \exp(\langle \tilde{\mathbf{q}}_n^{(l)}, \tilde{\mathbf{k}}^{(l)m} \rangle / \sqrt{D}) \tilde{\mathbf{v}}^{(l)m} + \sum_{i=1}^n \exp(\langle \mathbf{q}_n^{(l)}, \mathbf{k}^{(l)i} \rangle / \sqrt{D}) \mathbf{v}^{(l)i}}{\sum_{m=1}^M \exp(\langle \tilde{\mathbf{q}}_n^{(l)}, \tilde{\mathbf{k}}^{(l)m} \rangle / \sqrt{D}) + \sum_{i=1}^n \exp(\langle \mathbf{q}_n^{(l)}, \mathbf{k}^{(l)i} \rangle / \sqrt{D})}, \tag{2}$$

where $\tilde{\mathbf{q}}_n^{(l)} = \tilde{\mathbf{W}}_Q^{(l)} \boldsymbol{x}_n^{(l)}$, $\tilde{\mathbf{k}}^{(l)m} = \tilde{\mathbf{W}}_K^{(l)} \boldsymbol{k}^{(l)m}$, $\tilde{\mathbf{v}}^{(l)m} = \tilde{\mathbf{W}}_V^{(l)} \boldsymbol{v}^{(l)m}$ denote the Q-K-V embedding vectors after projection of the KB part. $\tilde{\mathbf{W}}_Q^{(l)} \in \mathbb{R}^{D \times D}, \tilde{\mathbf{W}}_K^{(l)}, \tilde{\mathbf{W}}_V^{(l)} \in \mathbb{R}^{D \times D_E}$ are the specific projection heads for the Q-K-V vectors of the KB. This rectangular attention have a time complexity of $\mathcal{O}((M + N) \cdot N \cdot D)$ and memory complexity of $\mathcal{O}((M + N) \cdot (N + D))$. Its computational overhead would grow linearly with $M$, which is more efficient than standard self-attention.

However, as the size of KB $M$ scales up heavily, the linearly growing time and memory complexity would still be a critical problem. AtlasKV can further improve the scalability of the KG augmented LLM with $\mathcal{O}\left((C_t \sqrt[3]{M} + N) \cdot N \cdot D\right)$ time complexity and $\mathcal{O}\left((C_m \sqrt[3]{M} + N) \cdot (N + D)\right)$ memory complexity, where $C_t$ and $C_m$ are constants that much smaller than $M$.

## 4 METHODOLOGY

***Overview of AtlasKV.*** Augmenting LLMs with super large and complex external knowledge, e.g. general KGs with billions of triples, often struggles with low generalization abilities and unbearable computational and memory overhead (Wang et al., 2024; Zhang et al., 2024b; Jin et al., 2024). To overcome these fundamental challenges, we propose **AtlasKV**, a scalable, effective and general method to integrate massive KGs into LLMs through two key innovations: (1) **KG2KV**, a new KG integration paradigm that naturally converts KG triples into Q-K-V data, enabling LLMs to achieve both enhanced generalization performance and efficient knowledge integration, and (2) **HiKVP**, a hierarchical key-value pruning algorithm that dramatically reduces computational and memory overhead while maintaining high knowledge grounding accuracy during inference time.

### 4.1 KG TO KV

Building on the observation that every triple in KGs can be naturally decomposed into Q-K-V strings (Verga et al., 2020), which shares very similar structure with the Q-K-V vectors of self-attention networks in LLMs, we introduce the concept of KGKV and employ the KG2KV pipeline to transform each KG triple into Q-K-V strings and their corresponding sentence embedding vectors.

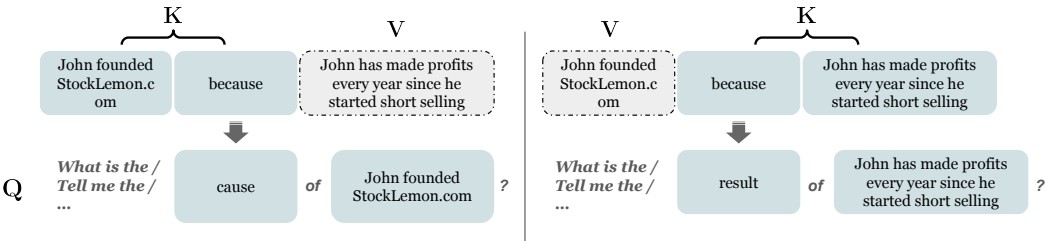

Figure 2: An example of how we transform the KG triples to Q-K-V data.

For a given KG triple $(h, r, t)$, we firstly mask its head $h$ or tail entity $t$ (could be named entity, event or concept entity). Then the masked entity can be the value we need in this triple. And then we will rewrite $r$ into a noun words according to the position we masked, which could be considered as the attribute of the entity that is not masked. For example, as shown in Figure 2, if we mask the tail entity in the triple, the relation "*because*" can be directly rewritten into its noun word "*cause*" through LLMs. And the key string of this tail-masked triple can be represented as "*the cause of John founded StockLemon.com*". If we mask the head entity in the triple, we need to rewrite the relation into its reversed noun word "*result*". And the key string of this head-masked triple can be represented as "*the result of John has made profits ...*". In this KG2KV pipline, we consider the masked entity as the value data, and the other entity as well as the relation as the key data, which complete KGKV data. Note that for the training data, we usually select named entities as the key to mask, and select event entities and relations as the value due to the reasons elaborated in Section 5.3. Then through a sentence encoder, KGKVs can be compressed and encoded into sentence embeddings $\mathbf{k}^m, \mathbf{v}^m$ to integrate to the attention layers of LLMs.

We also need the query sentence of each triple to serve as the training data. The query string can be obtained by adding various questioning prefix to the key strings. For example, the questioning prefix can be "*What is ...*", "*Tell me ...*", or "*Provide details on ..*". This design can ensure the model would not overfit to a specific type of questioning way.

Compared to directly synthesizing Q-K-V data from documents with limited human-defined schemas, the massive relations from KGs can guarantee the diversity of the enquiry attributes in the constructed training data with KG2KV method. Besides, due to the only strings we need input into the LLMs in KG2KV is the masked position and the relation, KG2KV needs fewer

Table 1: Comparison of the data diversity ratio and average token cost between KG2KV and synthetic method.

| Method | Diversity Ratio ↑ | Avg. Token Cost ↓ |
|---|---|---|
| Synthetic | 0.003% | 349.9 |
| KG2KV | **7.864%** | **165.7** |

token cost than directly synthetic method. We also provide the prompt template in Appendix H. As shown in Table 1, KG2KV holds a significantly higer diversity ratio (number of unique enquiry attributes divided by the total number of triples) of 7.864% and a lower average token cost of 165.7 than the synthetic method. We also provide some samples of the training data constructed by synthetic and KG2KV methods respectively in Appendix G.1. And the influence of relation rewriting process, which is the only part that consumes tokens in KG2KV, is also analyzed in Appendix B.2. As verified, more accurate key strings with relation rewriting in KG2KV are more beneficial than the simple combinations of relation and unmasked entity strings without relation rewriting.

## 4.2 AUGMENTING LLM WITH KGKVS

In this section, we will introduce how AtlasKV integrates the KGKVs constructed in the previous section into the LLMs with the help of hierarchical key-value pruning (HiKVP), which makes it more scalable than other methods.

***Hierarchical Clustering on KGKVs.*** Inspired by various works (Sarthi et al., 2024; Zhang et al., 2024a; Huang et al., 2025; Zhang et al., 2025b) that achieve some success by organizing hierarchical knowledge structure on textual chunks, which is an intuitive way to organize the world's knowledge, we employ the hierarchical clustering to cluster the keys of KGKVs into a hierarchical structure. This design aims to share the computational and memory burden during inference time on each layer of the hierarchical knowledge keys. Specifically, as previous works (Sarthi et al., 2024; Huang et al.,

2025) did, we first employ Uniform Manifold Approximation and Projection (UMAP) to reduce the dimension of the knowledge keys, and then employ Gaussian Mixture Models (GMMs) to cluster the knowledge keys into a hierarchical structure. Each key vector in a higer layer is the pooling of the key vectors in the lower layer. In AtlasKV, we set the number of layers to be 3, which can also be larger according to the actual situation. We select 3 layers because that is the minimum number of layers to include all of the definitions we need in AtlasKV. To share the computational and memory burden equally, we set the size of clusters in each layer to be the same, which is $S = \left\lceil \sqrt[3]{M} \right\rceil$. Then we can have the base embeddings of three layer knowledge keys $\boldsymbol{k}_L^m \in \mathbb{R}^{M_L \times D_E}$, $\boldsymbol{k}_I^m \in \mathbb{R}^{M_I \times D_E}$, $\boldsymbol{k}_R^m \in \mathbb{R}^{M_R \times D_E}$, where $M_L = M, M_I = \left\lceil M^{\frac{2}{3}} \right\rceil, M_R = \left\lceil \left\lceil M^{\frac{2}{3}} \right\rceil M^{-\frac{1}{3}} \right\rceil$. And $D_E$ is the embedding dimension of the sentence encoder.

***Knowledge Augmentation.*** During the tuning process of AtlasKV, we do not need to prune the KGKV pairs. Because due to the generalization capability of AtlasKV as verified in Section 5, we do not need such large scale KGKVs in the tuning process. And we use an equivalent attention method to replace the rectangular attention in KBLaM for knowledge augmentation. At the $l$-th attention layer and $n$-th token, it is computed as

$$\tilde{\mathbf{y}}_n^{(l)} = \lambda_{kg} \cdot \text{Softmax}\left(\text{logits}_{kg_L}\right) \cdot \tilde{\mathbf{v}}^{(l)} + \lambda_{seq} \cdot \text{Softmax}\left(\text{logits}_{seq}\right) \cdot \mathbf{v}^{(l)}, \tag{3}$$

where $\text{logits}_{kg_L}$ is the KG part of the attention output, $\text{logits}_{seq}$ is the sequence part of the attention output. For the weights $\lambda_{kg}$ and $\lambda_{seq}$ of these two softmax results, we have

$$\lambda_{kg} = \frac{\sum_{i=1}^{M} \exp(\text{logits}_{kg_L}^i)}{\sum_{i=1}^{M} \exp(\text{logits}_{kg_L}^i) + \sum_{i=1}^{n} \exp(\text{logits}_{seq}^i)}, \tag{4}$$

$$\lambda_{seq} = \frac{\sum_{i=1}^{n} \exp(\text{logits}_{seq}^i)}{\sum_{i=1}^{M} \exp(\text{logits}_{kg_L}^i) + \sum_{i=1}^{n} \exp(\text{logits}_{seq}^i)}. \tag{5}$$

And we also have

$$\text{logits}_{kg_L}^i = \langle \tilde{\mathbf{q}}_n^{(l)}, \tilde{\mathbf{k}}_L^{(l)i} \rangle / \sqrt{D}, \text{logits}_{seq}^i = \langle \mathbf{q}_n^{(l)}, \mathbf{k}^{(l)i} \rangle / \sqrt{D}, \tag{6}$$

where $\tilde{\mathbf{k}}_L^{(l)i} = \tilde{\mathbf{W}}_K^{(l)} \boldsymbol{k}_L^i$ and $\mathbf{k}^{(l)i} = \mathbf{W}_K^{(l)} \boldsymbol{k}^i$. The only learnable variables $\theta$ are the KG-specific query heads $\tilde{\mathbf{W}}_Q$ and KG projection heads $\tilde{\mathbf{W}}_K, \tilde{\mathbf{W}}_V$. Then we optimize $\theta$ using LLMs' original auto-regressive training objective:

$$p(v|\mathcal{M}, q) = \prod_{i=1}^{L} p_\theta(x_i|\mathcal{M}, q_{<i}, v_{<i}), \tag{7}$$

where $L$ is the total length of the query $q$ and the answer $v$. And $i$ denotes the $i$-th token in the combination of $q$ and $v$. $q_{<i}$ and $v_{<i}$ are the query and answer tokens in all turns before the current prediction token $x_i$. It is also very intuitive that the attention output at both the prefilling process and each decoding step is a weighted combination of the KG part and the sequence part values. And the weights are determined dynamically according to the logits of the KG and sequence part. The equivalence to the rectangular attention can be proven in Appendix C.

***Hierarchical Key-Value Pruning.*** During inference time, usually there are only a few relevant KG values needed for the query. And the injection of irrelevant knowledge would also introduce noise. So we use a hierarchical key-value pruning (HiKVP) pipeline as shown in Figure 3 to efficiently and scalably find the most relevant KGKVs for the query. And the attention will be computed as

$$\tilde{\mathbf{y}}_n^{(l)} = \bar{\lambda}_{kg} \cdot \text{Softmax}\left(\bar{\text{logits}}_{kg}\right) \cdot \bar{\tilde{\mathbf{v}}}^{(l)} + \lambda_{seq} \cdot \text{Softmax}\left(\text{logits}_{seq}\right) \cdot \mathbf{v}^{(l)}, \tag{8}$$

where the $\bar{\text{logits}}_{kg}$ and $\bar{\tilde{\mathbf{v}}}^{(l)}$ are the pruned KG logits and values in the leaf-layer. $\bar{\lambda}_{kg}$ is the weight based on the pruned KG logits. They can be obtained by the following steps.

**Step 1.** Initially, only the root-layer projected key vectors $\tilde{\mathbf{k}}_R^{(l)}$ are uploaded in the GPU memory, while other layer key and value vectors ($\tilde{\mathbf{k}}_I^{(l)}, \tilde{\mathbf{k}}_L^{(l)}$, and $\tilde{\mathbf{v}}^{(l)}$) remain in the CPU memory. We first compute attention weights between the query and root-layer keys:

$$\text{logits}_{kg_R} = \langle \tilde{\mathbf{q}}_n^{(l)}, \tilde{\mathbf{k}}_R^{(l)} \rangle / \sqrt{D}. \tag{9}$$

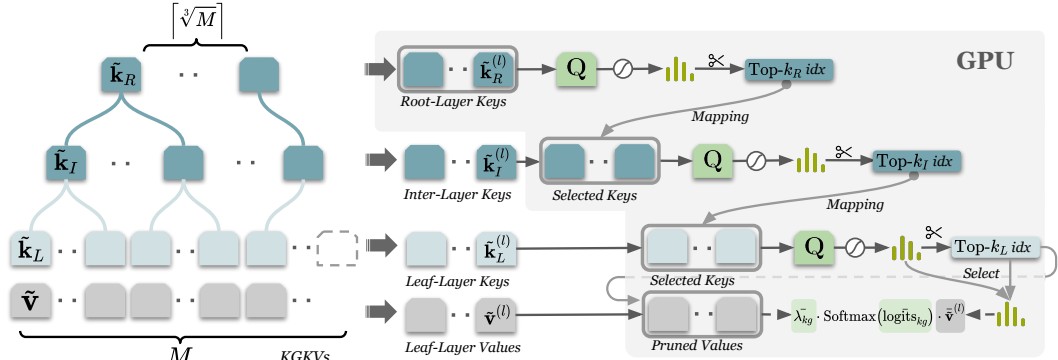

Figure 3: An overview of hierarchical key-value pruning (HiKVP) with three layers of knowledge keys at the $l$-th attention layer. The gray background indicate that the part is stored and computed in the GPU memory.

After calculating the softmax of it, we prune root keys by selecting the keys with top-$k_R$ highest scores and use a mapping to obtain the included inter-layer keys $\bar{\tilde{\mathbf{k}}}_I^{(l)} \in \mathbb{R}^{(k_R S) \times D}$. And the root-layer keys will be offloaded back to the CPU memory.

**Step 2.** We conduct the similar process to the step 1, but with the selected inter-layer keys. We upload the selected inter-layer keys to the GPU memory, and compute the attention weights:

$$\bar{\text{logits}}_{kg_I} = \langle \tilde{\mathbf{q}}_n^{(l)}, \bar{\tilde{\mathbf{k}}}_I^{(l)} \rangle / \sqrt{D}. \tag{10}$$

Then we can prune the selected inter-layer keys and obtain the selected leaf-layer keys $\bar{\tilde{\mathbf{k}}}_L^{(l)} \in \mathbb{R}^{(k_I S) \times D}$ in the same way. And the selected inter-layer keys will be offloaded to the CPU memory.

**Step 3.** Finally, we upload the selected leaf-layer keys from the CPU to GPU memory. Then we compute the attention weights after the softmax calculation and prune the leaf-layer keys by directly selecting the corresponding logits with top-$k_L$ highest softmax scores:

$$\bar{\text{logits}}_{kg} = \text{TopK\_logits}\left(\text{Softmax}\left(\bar{\text{logits}}_{kg_L}\right), k_L\right) \in \mathbb{R}^{k_L \times 1}, \bar{\text{logits}}_{kg_L} = \langle \tilde{\mathbf{q}}_n^{(l)}, \bar{\tilde{\mathbf{k}}}_L^{(l)} \rangle / \sqrt{D}. \tag{11}$$

And through the mapping indices, we can also obtain the pruned values $\bar{\tilde{\mathbf{v}}}^{(l)} \in \mathbb{R}^{k_L \times D}$ and upload them to the GPU memory. Then $\bar{\text{logits}}_{kg}, \bar{\tilde{\mathbf{v}}}^{(l)}$ and $\bar{\lambda}_{kg}$ can be obtained and the attention output of Equation 8 during the inference time can be finally computed.

All these steps of HiKVP can be done in $\mathcal{O}\left((C_t \sqrt[3]{M} + N) \cdot N \cdot D\right)$ time complexity and $\mathcal{O}\left((C_m \sqrt[3]{M} + N) \cdot (N + D)\right)$ memory complexity, respectively, where $C_t$ and $C_m$ are constants that are much smaller than $M$. We also provide the detailed derivation in Appendix D.

## 5 EXPERIMENTS

In this section, we report the performance of AtlasKV from GPU memory cost, knowledge grounding accuracy and the relevance of the generation results perspectives. And we also compare the performance of AtlasKV with other knowledge integration baseline methods to demonstrate the superiority of AtlasKV. We also report the training and evaluation details of AtlasKV in Appendix A.1 and Appendix A.2.

### 5.1 EXPERIMENTAL SETTINGS

**Baselines.** Following the settings in KBLaM (Wang et al., 2024), we compare AtlasKV with both non-parametric and parametric methods. For non-parametric methods, we include **in-context learning (ICL)**, which is the basic knowledge augmentation paradigm used in RAG methods. For parametric methods, we include **KBLaM**, which is an advanced parametric knowledge augmentation paradigm. We also include **zero-shot learning** to provide some boundaries of our experimental results.

**Training Datasets.** In AtlasKV, we construct the Q-K-V training dataset **ATLAS-Wiki-QKV** with **ATLAS-Wiki**, which is one of ATLAS (Bai et al., 2025) family KGs with 900+ million nodes and 5.9 billion edges containing both event and named entities. Because it offer sufficiently large KGs that enable us to train the model and comprehensively assess the performance of AtlasKV and other baseline methods across various KG sizes. And we also use the **Synthetic** training dataset used in KBLaM (Wang et al., 2024) to compare with.

**Evaluation Datasets.** To comprehensively assess the performance of AtlasKV and other baseline methods in a scenario closer to the real world, we test all methods in the OOD settings. We not only include the **Enron** dataset (Klimt & Yang, 2004), which is an OOD dataset used in KBLaM (Wang et al., 2024), but also introduce the **ATLAS-CC-QKV** dataset (Bai et al., 2025) and **ATLAS-Pes2o-QKV** dataset (Bai et al., 2025), which are also constructed from **ATLAS-CC** and **ATLAS-Pes2o** in ATLAS (Bai et al., 2025) with the KG2KV method, respectively, to evaluate the performance of different methods in a more comprehensive way. Note that ATLAS-CC-QKV and ATLAS-Pes2o-QKV are much harder than Enron dataset because they are constructed from more complex KGs and include much more unique enquiry attributes that are closer to the real world scenarios. With these OOD datasets, we can better evaluate the generalization capabilities of various methods.

## 5.2 EXPERIMENTAL RESULTS

| Method | Time Complexity | Memory Complexity |
|--------|-----------------|-------------------|
| ICL | $\mathcal{O}\left((MT+N)^2 \cdot D\right)$ | $\mathcal{O}\left((MT+N)\cdot(MT+N+D)\right)$ |
| RAG | $\mathcal{O}\left(M+RT+(RT+N)^2 \cdot D\right)$ | $\mathcal{O}\left((RT+N)\cdot(RT+N+D)\right)$ |
| CAG | $\mathcal{O}\left((RT+N)^2 \cdot D\right)$ | $\mathcal{O}\left((RT+N)\cdot(N+D)\right)$ |
| KBLaM | $\mathcal{O}\left((M+N)\cdot N \cdot D\right)$ | $\mathcal{O}\left((M+N)\cdot(N+D)\right)$ |
| AtlasKV | $\mathcal{O}\left((C_t\sqrt[3]{M}+N)\cdot N \cdot D\right)$ | $\mathcal{O}\left((C_m\sqrt[3]{M}+N)\cdot(N+D)\right)$ |

Table 2: Comparison of the time and memory complexity of AtlasKV, KBLaM, RAG, CAG (Chan et al., 2025) and ICL methods, where the parts marked in teal color represent they could be very large.

***AtlasKV is more scalable with HiKVP.*** To verify the scalability of AtlasKV with HiKVP, we compare the GPU memory usage at inference time of AtlasKV and other methods across a wide range of KG sizes from 1 to 1B triples. As shown in Figure 4, the colored areas represent how much VRAM is saved compared with the other method. It demon-

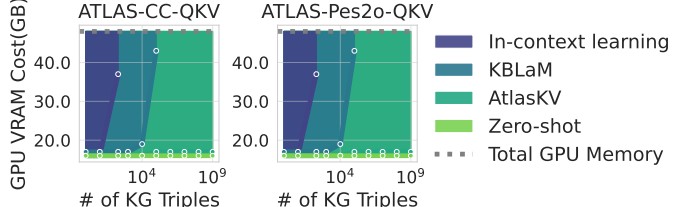

Figure 4: GPU memory usage comparison of AtlasKV and other methods across various KG sizes from 1 to 1B triples.

strates that with the help of HiKVP, AtlasKV can save a huge amount of VRAM compared with ICL as well as KBLaM and achieve a much lower GPU memory cost. With the increasing of KG scale, the VRAM usage of AtlasKV even just a little bit higher than the zero-shot generation. And in AtlaskV, less than 20GB VRAM is required to augment LLMs with 1B triples. However, in KBLaM, over 40GB VRAM is required to deal with even 100K triples. The key reason why AtlasKV can achieve this is that HiKVP significantly reduces the time and memory complexity of KBLaM from linear to sub-linear, as demonstrated in Table 2, where $T$ denotes the average token length of the triples. In ICL-based RAG methods, when the size of relevant triples $R$ scales up, the inference latency and VRAM usage caused by long context prior dependence would grow heavily. And their performance will also influenced by the lost-in-the-middle dilemma (Liu et al., 2023), which would not exist in AtlasKV. And compared with the increased cost with every additional retrieved $R$ in CAG (Chan et al., 2025), which is a new caching-based knowledge augmentation paradigm, AtlasKV trades the sub-linear attention-based retrieval cost for the ability to perform efficient inference independent of the scale of the retrieved relevant knowledge.

***AtlasKV is more accurate and generalizable with KGKVs.*** Not only can AtlasKV save a huge amount of inference cost with strong scalability, but it can also maintain a higher knowledge grounding performance with strong generalization ability. To quantitatively assess the knowledge grounding performance of AtlasKV, we extract the averaged-over-heads KG part post-softmax attention scores at the 15th layer due to the reason described in Appendix A.2, and then we can obtain the Top-1

| Method | Steps | $10^1$ Triples ACC@1 | ACC@5 | $10^2$ Triples ACC@1 | ACC@5 | $10^3$ Triples ACC@1 | ACC@5 | $10^4$ Triples ACC@1 | ACC@5 |
|---|---|---|---|---|---|---|---|---|---|
| *Eval on Enron* | | | | | | | | | |
| KBLaM | 3e3 | **100.0** | **100.0** | 50.9 | 76.4 | 29.1 | 56.4 | 9.1 | 20.0 |
| | 2e4 | 90.0 | **100.0** | 50.9 | 83.6 | 25.5 | 47.3 | 7.3 | 23.6 |
| AtlasKV (128-64-16) | 3e3 | **100.0** (+0.0) | **100.0** (+0.0) | 67.3 (+16.4) | 90.9 (+7.3) | 41.8 (+12.7) | 50.9 | 21.8 (+12.7) | 32.7 (+12.7) |
| AtlasKV w/o HiKVP | 3e3 | **100.0** (+0.0) | **100.0** (+0.0) | **76.4** (+25.5) | **92.7** (+9.1) | **56.4** (+27.3) | **80.0** (+23.6) | **27.3** (+18.2) | **47.3** (+27.3) |
| *Eval on ATLAS-Pes2o-QKV* | | | | | | | | | |
| KBLaM | 3e3 | 40.0 | 80.0 | 16.4 | 45.5 | 5.5 | 14.5 | 0.0 | 3.6 |
| | 2e4 | 50.0 | 80.0 | 25.5 | 52.7 | 3.6 | 14.5 | 0.0 | 5.5 |
| AtlasKV (128-64-16) | 3e3 | 90.0 (+40.0) | **100.0** (+20.0) | 87.3 (+61.8) | 92.7 (+40.0) | 52.7 (+47.2) | 70.9 (+56.4) | 16.4 (+16.4) | 49.0 (+43.5) |
| AtlasKV w/o HiKVP | 3e3 | **100.0** (+50.0) | **100.0** (+20.0) | **92.7** (+67.2) | **100.0** (+47.3) | **72.7** (+67.2) | **90.9** (+76.4) | **47.3** (+47.3) | **67.2** (+61.7) |
| *Eval on ATLAS-CC-QKV* | | | | | | | | | |
| KBLaM | 3e3 | 60.0 | 90.0 | 21.8 | 38.2 | 12.7 | 23.6 | 3.6 | 10.9 |
| | 2e4 | 50.0 | **100.0** | 23.6 | 56.4 | 10.9 | 21.8 | 3.6 | 10.9 |
| AtlasKV (128-64-16) | 3e3 | **100.0** (+0.0) | **100.0** (+0.0) | 89.1 (+65.5) | 90.9 (+34.5) | 61.8 (+49.1) | 74.5 (+50.9) | 40.0 (+36.4) | 54.5 (+43.6) |
| AtlasKV w/o HiKVP | 3e3 | **100.0** (+0.0) | **100.0** (+0.0) | **96.4** (+72.8) | **100.0** (+43.6) | **83.6** (+70.9) | **96.4** (+72.8) | **61.8** (+58.2) | **81.8** (+70.9) |

Table 3: The knowledge grounding performance of AtlasKV against KBLaM with all-MiniLM-L6-v2 as the sentence encoder on three OOD evaluation datasets across various tuning steps and KG sizes. We defaultly set the top-k in HiKVP to 128, 64, and 16 for the $k_R, k_I, k_L$ respectively.

and Top-5 accuracy of the knowledge grounding performance. As shown in Table 3, across all three OOD datasets and a wide range of KG sizes, AtlasKV achieves significantly higher Top-1 and Top-5 accuracy than KBLaM with the KGKVs as the training data. Especially in ATLAS-Pes2o-QKV and ATLAS-CC-QKV datasets, which are much harder due to their complex and diverse enquiry attributes, KBLaM performs very bad because there are too limited enquiry attributes in Synthetic training data to make it more generalizable. However, only 20K KGKV samples as the training data are needed to make AtalsKV much more accurate and generalizable. It also suggests KGKVs can make the training process more efficient with only 3K steps, compared with the 20K training steps reported in KBLaM. We also experiment with different top-k settings in HiKVP in Appendix B.4.1. Another interesting observation is that, even with HiKVP, there is not a big performance drop of AtlasKV and it still performs better than KBLaM. This is mainly because the specific heads trained in AtlasKV have the capabilities to conduct fuzzy retrieval at different layers of semantic granularity. Besides, we observe the training dynamics of AtlasKV in Appendix E, which is that **from a specific training step, the model regularly start learning to retrieve relevant knowledge from the external KG triples, instead of brute force over-fitting**. We also report the results with a larger model as the sentence encoder in Appendix B.1.

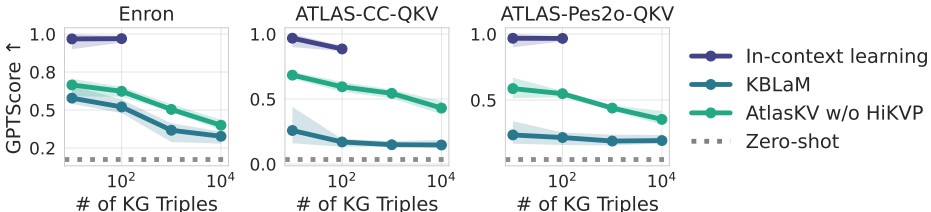

Figure 5: Scored by GPT-4o between 0 and 1, the shaded area exhibits the standard error over 5 random seeds. The score of each random seed is also the average of 5 generation results.

We also use GPT-4o (Hurst et al., 2024) to score the relevance between the ground truth and generated answers. As shown in Figure 5, AtlasKV achieves significantly higher GPTScores than KBLaM. Although ICL can generate a very accurate result with over 0.9 scores, it is super time-consuming to put all external knowledge into the context of LLMs. Expecially when there are more than 100 triples in a KG, over 48GB VRAM is required and can not be run on the limited GPU memory. And KBLaM also performs poorly on the two difficult datasets, which is also due to the reasons we explained above. **Remarkably, despite having only limited training samples with enquiry attributes similar to Enron in ATLAS-Wiki-QKV, AtlasKV still outperforms KBLaM on both knowledge grounding accuracy and answer relevance metrics, even though KBLaM's training data contains exactly the same enquiry attributes as Enron.** This is mainly due the the diversity of the enquiry attributes in ATLAS-Wiki-QKV, which is constructed by KG2KV module. We have compared that with fully synthetic method in Table 1 and it makes AtlasKV have the capability to be generalized to more unseen enquiry attributes in complex scenarios. We also provide some samples of ATLAS-Wiki-QKV and Synthetic dataset in Appendix G.1.

## 5.3 ABLATION STUDY

| Method | Steps | $10^1$ Triples | | $10^2$ Triples | | $10^3$ Triples | | $10^4$ Triples | |
|---|---|---|---|---|---|---|---|---|---|
| | | ACC@1 | ACC@5 | ACC@1 | ACC@5 | ACC@1 | ACC@5 | ACC@1 | ACC@5 |
| *Eval on ATLAS-Pes2o-QKV* | | | | | | | | | |
| AtlasKV w/o HiKVP | 3e3 | **100.0** | **100.0** | **92.7** | **100.0** | **72.7** | **90.9** | **47.3** | **67.2** |
| AtlasKV w/o HiKVP & Event | 3e3 | 90.0 | **100.0** | 80.0 | 89.1 | 34.5 | 63.6 | 9.1 | 36.4 |
| AtlasKV w/o HiKVP & Entity | 3e3 | **100.0** | **100.0** | 49.0 | 67.3 | 20.0 | 30.9 | 3.6 | 5.5 |
| *Eval on Enron* | | | | | | | | | |
| AtlasKV w/o HiKVP | 3e3 | **100.0** | **100.0** | **76.4** | **92.7** | **56.4** | **80.0** | **27.3** | **47.3** |
| AtlasKV w/o HiKVP & Event | 3e3 | 80.0 | **100.0** | 73.6 | 84.5 | 48.0 | 66.0 | 10.9 | 38.2 |
| AtlasKV w/o HiKVP & Entity | 3e3 | 40.0 | **100.0** | 40.0 | 74.5 | 16.4 | 27.3 | 1.8 | 9.1 |

Table 4: The knowledge grounding performance of different variants of AtlasKV with all-MiniLM-L6-v2 as the sentence encoder on three OOD evaluation datasets across various tuning steps and KG sizes.

***Cooperating named and event entities together in KG2KV process helps with the model's learning.*** To analyze the reasonability of the design described in Section 4.1, we conduct experiments on the variant with only KG2KV component, which are denoted as "AtalsKV w/o HiKVP". Because we need focus on the ablations of training data and avoid the influence of pruning process. We compare the training data containing both named and event entities with the variants containing only named or event entities, which are denoted as "AtalsKV w/o HiKVP & Event" and "AtalsKV w/o HiKVP & Entity". Here is the analysis based on the results in Table 4: (1) Without any one kind of entities, there will be a drop of the knowledge grounding performance. (2) Especially when there are only event entities, due to the high complex semantics in both key and value strings, it becomes very hard for the specific heads to learn from scratch, which leads to a huge performance drop. (3) When only named entities are employed in KG2KV, the performance drop is smaller because the key and value strings of them are shorter and the semantics are simpler, which makes it easier for the specific heads to learn from scratch. But the worse performance with only named entities suggests that we still need some complex semantics in event entities to help the model to learn better.

## 6 CONCLUSION

In this paper, we presented AtlasKV, a scalable, effective, and general framework to augment LLMs with billion-scale knowledge graphs under very low GPU memory budgets. Compared with non-parametric methods, AtlasKV requires no external retrievers and does not depend on long context prior, which could lead to substantial inference latency. Compared with traditional parametric methods, AtlasKV can be adapted to new knowledge in a training-free manner. We achieve that by (1) KG2KV which naturally converts KG triples into Q-K-V data, enabling LLMs to achieve both enhanced generalization performance and efficient knowledge integration, and (2) HiKVP which conducts hierarchical key-value pruning to dramatically reduces computational and memory costs while maintaining high performance during inference time.

## 7 ETHICS STATEMENT

We affirm adherence to the ICLR Code of Ethics. This work develops AtlasKV does not involve human subjects or interventions. We use publicly available datasets (e.g., ATLAS family KGs, Synthetic, and Enron datasets) under their licenses. We construct Q-K-V data via KG2KV without collecting new personal data or attempting deanonymization. We also comply with model or API providers' terms (e.g., LLaMA3.1-8B-Instruct, GPT-4o, and GPT-4o-mini) without uploading proprietary or sensitive information. No undisclosed conflicts of interest exist. All experiments were performed on a single 48GB GPU, and we encourage energy-efficient configurations.

## 8 REPRODUCIBILITY STATEMENT

**LLMs for Training and Evaluation.** Three LLMs are used in our AtlasKV experiments: LLaMA3.1-8B-Instruct, GPT-4o, and GPT-4o-mini. We use LLaMA3.1-8B-Instruct as the backbone of AtlasKV and it can be obtained at Hugging Face. GPT-4o and GPT-4o-mini are used to score the generated

answers and rewrite the relations in KG2KV process, respectively. They can be accessed via OpenAI API calls.

**Training and Evaluation Details.** We provide comprehensive descriptions about our training and evaluation settings in Appendix A, including the hyper-parameter settings and detailed processing steps. All of the datasets we used in our work can be obtained through public resources (Bai et al., 2025; Wang et al., 2024; Klimt & Yang, 2004).

## 9 ACKNOWLEDGMENTS

The authors of this paper were supported by the ITSP Platform Research Project (ITS/189/23FP) from ITC of Hong Kong, SAR, China, and the AoE (AoE/E-601/24-N), and the GRF (16205322) from RGC of Hong Kong, SAR, China, and the support of RGC JRFS2526-6S10. We also thank the support from Huawei.

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

# Appendix of `AtlasKV`

## CONTENTS

# A    TRAINING AND EVALUATION DETAILS OF ATLASKV

For the reproducibility of our work, we provide the training and evaluation details of AtlasKV in this section. All training and evaluation experiments are conducted in a single 48GB GPU under bfloat16 precision.

## A.1    TRAINING SETTINGS

We use the same training settings and methods to construct training samples as in KBLaM (Wang et al., 2024), where we also use the instruction fine-tuned version of LLaMA3.1-8B (Dubey et al., 2024), which is represented as LLaMA3.1-8B-Instruct. As an essential part in AtalsKV, any sentence encoder can be employed in our method to compute the base key and value embeddings. We conduct experiments with open-source all-MiniLM-L6-v2 (Wang et al., 2020b) ($D_E = 384$) and closed-source text-embedding-3-large ($D_E = 3072$) through API respectively to serve as the sentence encoders.

To initialize the parameters we need to train in AtlasKV, the KG-specific query heads $\tilde{\mathbf{W}}_Q^{(l)}$ are initialized from $\mathbf{W}_Q^{(l)}$ at each layer and the KG projection heads $\tilde{\mathbf{W}}_K, \tilde{\mathbf{W}}_V$ are initialized randomly. We sample and select only 20K triples from ATLAS-Wiki, which contains 1.492B triples, to construct the Q-K-V training dataset. We use AdamW (Loshchilov & Hutter, 2017) as the optimizer with a step size of $1 \times 10^{-3}$ and a cosine learning rate decay to $1 \times 10^{-5}$ for 3K iterations. Each iteration consists a batch size of 10. And the KG sizes at each iteration will increase 4 every 100 iterations. The reason why we only need a small number of training steps (KG sizes for training) is that we found AtlasKV also exhibits some generalization capabilities across various KG sizes as verified in Section 5.2. For example, within 3K iterations, even though the maximum size of KG that is used to train the AtlasKV is only 120. However, it can already perform well with larger scale of KGs. And we integrate the KGKVs into the LLM's attention every 3 layers for efficient training and inference.

Note that all of the base embeddings of the KGKVs are computed offline. So during both of the training and inferencing processes, we only need to load them from hard disk and project some of them into the embedding space of LLMs.

## A.2    EVALUATION SETTINGS

In the knowledge grounding experiments, to verify AtlasKV can exhibit superior knowledge grounding accuracy even with small sentence encoders, we employ a lightweight open-source sentence transformer (Reimers & Gurevych, 2019) all-MiniLM-L6-v2 (Wang et al., 2020b) [1] to serve as the sentence encoder here. In the generation relevance experiments, we need stronger sentence encoder to let the value embeddings in KGKV2 have enough semantics. So in these experiments, we select a bigger OpenAI sentence encoder text-embedding-3-large through API. And we also demonstrate in Appendix B.1 that with text-embedding-3-large as the encoder, AtlasKV can also achieve a higher knowledge grounding accuracy.

For knowledge grounding performance evaluation, we extract the averaged-over-heads attention scores of the KG parts that are computed by softmax at the 15th attention layer (for LLaMA3.1-8B-Instruct that has attention layers from 0-31). We did that due to this attention layer is mainly responsible to retreive the accurate external knowledge keys and the external knowledge key embeddings after this attention layer show a higher degree of variation, which has been verified in KBLaM (Wang et al., 2024).

For generation relevance evaluation, like many previous works did (Edge et al., 2024; Huang et al., 2025; Guo et al., 2024; Es et al., 2024), we employ GPT-4o as the evaluator to score the relevance between the generated results and ground truth answers. And the prompt template is shown in Figure 11. To make that statistically significant, we run 5 random seeds for each experiment and we also generate the score 5 times for each seed to get the average score.

---

[1]https://huggingface.co/sentence-transformers/all-MiniLM-L6-v2

# B    EXTENDED EXPERIMENTS

## B.1    KNOWLEDGE GROUNDING WITH DIFFERENT ENCODERS

In this section, we conduct experiments with text-embedding-3-large as the sentence encoder to verify that AtlasKV can achieve superior knowledge grounding accuracy with different sentence encoders. Because the output dimension of text-embedding-3-large is 3072, which is much larger than the output dimension of all-MiniLM-L6-v2 (384), we increase the training steps of AtlasKV from 3K to 10K to make sure the training process can well converge. As shown in Table 5, with text-embedding-3-large as the sentence encoder, AtlasKV can still achieve a higher knowledge grounding accuracy than KBLaM at most of the cases. It further demonstrates the adaptivities of AtlasKV to various sentence encoders.

| Method | Steps | $10^1$ Triples | | $10^2$ Triples | | $10^3$ Triples | | $10^4$ Triples | |
|---|---|---|---|---|---|---|---|---|---|
| | | ACC@1 | ACC@5 | ACC@1 | ACC@5 | ACC@1 | ACC@5 | ACC@1 | ACC@5 |
| *Eval on Enron* | | | | | | | | | |
| KBLaM | 1e4 | 80.0 | **100.0** | 31.0 | 69.0 | 32.0 | 56.0 | 20.7 | 25.9 |
| AtlasKV (128-64-16) | 1e4 | **100.0** | **100.0** | 77.6 | 89.7 | 36.2 | 50.0 | 19.0 | 25.9 |
| AtlasKV w/o HiKVP | 1e4 | **100.0** | **100.0** | **86.2** | **96.6** | **62.1** | **84.5** | **36.2** | **46.6** |
| *Eval on ATLAS-Pes2o-QKV* | | | | | | | | | |
| KBLaM | 1e4 | 60.0 | 90.0 | 20.7 | 56.9 | 13.8 | 34.5 | 6.9 | 15.5 |
| AtlasKV (128-64-16) | 1e4 | **100.0** | **100.0** | 93.0 | 98.2 | 49.1 | 63.2 | 28.1 | 43.9 |
| AtlasKV w/o HiKVP | 1e4 | **100.0** | **100.0** | **96.5** | **100.0** | **71.9** | **91.2** | **36.8** | **59.6** |
| *Eval on ATLAS-CC-QKV* | | | | | | | | | |
| KBLaM | 1e4 | 80.0 | **100.0** | 43.9 | 73.7 | 17.5 | 35.1 | 10.5 | 12.8 |
| AtlasKV (128-64-16) | 1e4 | **100.0** | **100.0** | 85.5 | 96.4 | 54.5 | 74.5 | 52.7 | 65.5 |
| AtlasKV w/o HiKVP | 1e4 | **100.0** | **100.0** | **85.5** | **98.2** | **80.0** | **92.7** | **65.5** | **81.8** |

Table 5: The knowledge grounding accuracy of AtlasKV against KBLaM with text-embedding-3-large as the sentence encoder across various tuning steps and KG sizes.

## B.2    INFLUENCE OF RELATION REWRITING IN KG2KV

Even though we have verified in Section 4.1 and Table 1 that compared with the fully Synthetic method, KG2KV is not only more effective but also cheaper in terms of the average token cost of processing each triple, the relation rewriting process in KG2KV will still introduce high token cost if the KG scale is very large. So we conduct experiments to analyze the sensitivity of AtlasKV to the quality of relation rewriting in KG2KV from two perspectives: (1) training data, (2) testing data.

For training data, we train the model on a naive version of ATLAS-Wiki-QKV (denoted as AtlasKV-Ntrain), in which we remove the relation rewriting process and directly use the combination of original relations and unmasked entities as the keys. Then we compare the knowledge grounding performance of AtlasKV-Ntrain and AtlasKV on the original version of ATLAS-Pes2o-QKV. We use all-MiniLM-L6-v2 as the sentence encoder. As shown in Table 6, without relation rewriting in the training data, although there is a slight performance drop in AtlasKV-Ntrain w/o HiKVP and AtlasKV-Ntrain, our model can still maintain a high knowledge grounding accuracy.

| Method | Steps | $10^1$ Triples | | $10^2$ Triples | | $10^3$ Triples | | $10^4$ Triples | |
|---|---|---|---|---|---|---|---|---|---|
| | | ACC@1 | ACC@5 | ACC@1 | ACC@5 | ACC@1 | ACC@5 | ACC@1 | ACC@5 |
| *Eval on ATLAS-Pes2o-QKV* | | | | | | | | | |
| KBLaM | 2e4 | 50.0 | 80.0 | 25.5 | 52.7 | 3.6 | 14.5 | 0.0 | 5.5 |
| AtlasKV (128-64-16) | 3e3 | 90.0 | **100.0** | 87.3 | 92.7 | 52.7 | 70.9 | 16.4 | 49.0 |
| AtlasKV-Ntrain (128-64-16) | 3e3 | 80.0 | **100.0** | 71.2 | 86.5 | 42.3 | 57.7 | 21.2 | 34.6 |
| AtlasKV w/o HiKVP | 3e3 | **100.0** | **100.0** | **92.7** | **100.0** | **72.7** | **90.9** | **47.3** | **67.2** |
| AtlasKV-Ntrain w/o HiKVP | 3e3 | 80.0 | **100.0** | 80.8 | 94.2 | 55.8 | 82.7 | 38.5 | 57.7 |

Table 6: Knowledge grounding performance comparison between AtlasKV-Ntrain and AtlasKV with all-MiniLM-L6-v2 as the sentence encoder across various KG sizes.

For testing data, we use the model trained on the original version of ATLAS-Wiki-QKV. And then we apply the above naive KG2KV process (without relation rewriting) to ATLAS-Pes2o-QKV, which is our testing data. We represent this version of AtlasKV as AtlasKV-Ntest. We still use all-MiniLM-L6-v2 as the sentence encoder. As shown in Table 7, without relation rewriting in testing data,

AtlasKV-Ntest w/o HiKVP and AtlasKV-Ntest can still perform well. The performance drops are also acceptable.

| Method | Steps | $10^1$ Triples ACC@1 | $10^1$ Triples ACC@5 | $10^2$ Triples ACC@1 | $10^2$ Triples ACC@5 | $10^3$ Triples ACC@1 | $10^3$ Triples ACC@5 | $10^4$ Triples ACC@1 | $10^4$ Triples ACC@5 |
|---|---|---|---|---|---|---|---|---|---|
| *Eval on ATLAS-Pes2o-QKV* | | | | | | | | | |
| KBLaM | 2e4 | 50.0 | 80.0 | 25.5 | 52.7 | 3.6 | 14.5 | 0.0 | 5.5 |
| AtlasKV (128-64-16) | 3e3 | 90.0 | 100.0 | 87.3 | 92.7 | 52.7 | 70.9 | 16.4 | 49.0 |
| AtlasKV-Ntest (128-64-16) | 3e3 | 90.0 | 100.0 | 86.5 | 90.4 | 46.2 | 61.5 | 23.1 | 36.5 |
| AtlasKV w/o HiKVP | 3e3 | 100.0 | 100.0 | 92.7 | 100.0 | 72.7 | 90.9 | 47.3 | 67.2 |
| AtlasKV-Ntest w/o HiKVP | 3e3 | 100.0 | 100.0 | 76.9 | 98.0 | 55.8 | 78.8 | 26.9 | 46.2 |

Table 7: Knowledge grounding performance comparison between AtlasKV-Ntest and AtlasKV with all-MiniLM-L6-v2 as the sentence encoder across various KG sizes.

The above results suggest that without relation rewriting process in training or testing data, which is the only process that needs LLMs in KG2KV, our model can still perform well. This is mainly because although more accurate key strings with relation rewriting in KG2KV are beneficial, simple combination of relation and unmasked entity strings can also be very helpful for distinguishing them. And the performance drops are also acceptable. This can also be considered as a tradeoff among accuracy, token consumption and KG scale.

### B.3 INFLUENCE OF KNOWLEDGE INJECTION FREQUENCY

In AtlasKV, similar to KBLaM (Wang et al., 2024), the knowledge injection frequency in self-attention layers of LLMs would also influence the knowledge grounding performance. We set the knowledge injection frequency $K = 3$ by default in all of our experiments. In this section, we will analyze the influence of knowledge injection frequency on the knowledge grounding accuracy of AtlasKV. Following the parameter sensitivity experiments in KBLaM, we set the knowledge injection frequency $K$ as 1, 3, 10 and then compare the knowledge grounding accuracy on the ATLAS-Pes2o-QKV dataset.

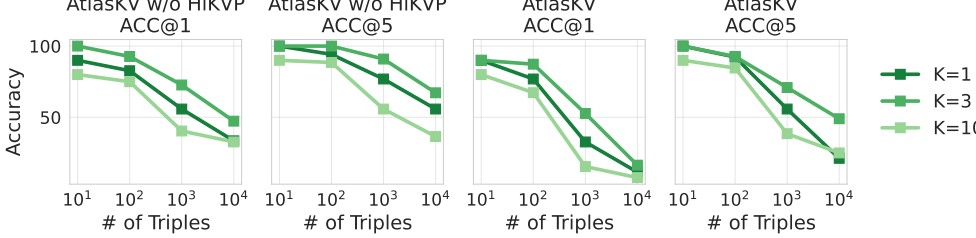

Figure 6: The knowledge grounding accuracy of AtlasKV and AtlasKV w/o HiKVP on ATLAS-Pes2o-QKV with various knowledge injection frequencies.

As shown in Table 6, in AtlasKV, too frequent or infrequent knowledge injection will both lead to suboptimal performance. $K = 3$ is the best choice for AtlasKV. This is mainly because too frequent knowledge injection will introduce noise in early attention layers and with lower frequency, the model will fail to ground accurate triples due to inadequate knowledge injection.

### B.4 ANALYSIS OF TOP-K SELECTION IN HIKVP

#### B.4.1 INFLUENCE OF VARIOUS TOP-K IN HIKVP

In the default settings of our previous experiments, we set the $k_R, k_I, k_L$ to 128, 64, and 16 respectively. In this section, we conduct experiments to investigate how different top-k settings at each layer of HiKVP influence the knowledge grounding accuracy. We test that based on our default settings, and we change one of $k_R, k_I, k_L$ to different values to see the influence of the top-k settings on the knowledge grounding accuracy of HiKVP. Specifically, we set $k_R, k_I, k_L$ to 128, 64, 32, 16, and 8, respectively, while keeping the other two layers the same as our default settings. And we set the candidate triples to $10^5$. As shown in Figure 7, we can see that the knowledge grounding accuracy of AtlasKV will be significantly improved if we increase $k_R$. And the performance will first improve and then slightly decrease when we increase $k_I$ or $k_L$. This suggests that the accurate retrieval ability

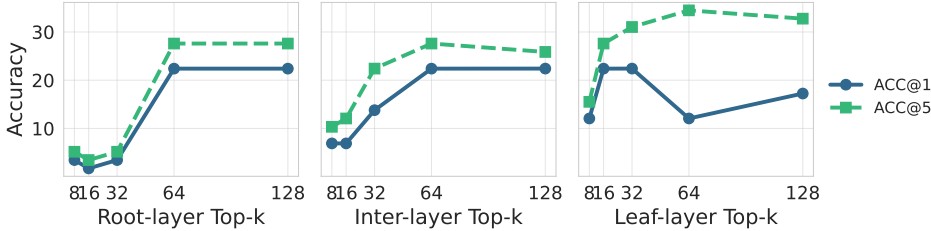

Figure 7: The knowledge grounding accuracy of AtlasKV on ATLAS-CC-QKV with different top-k settings at each layer.

of AtlasKV is stronger than the fuzzy retrieval ability of it. And the reason why too large $k_I$ or $k_L$ will hurt the performance might be that the noise candidate keys selected in early attention layers would influence the retrieval accuracy of the later attention layers.

### B.4.2 INTER-LEVEL CORRELATIONS OF TOP-K IN HIKVP

To find out the inter-level correlations in top-k selections of HiKVP, we compare the knowledge grounding accuracy at the inter-layer as well as root-layer with different top-k selections at a higher layer. To be specific, to find the correlations of top-k selections between the root-layer and inter-layer, we remove the pruning process in the leaf-layer and compare the knowledge grounding performance (ACC@5) with various $K_R$ and fixed $K_I = 8, 16, 32, 64, 128$ respectively. Similarly, to find the correlations of top-k selections between the inter-layer and leaf-layer, we compare the knowledge grounding performance (ACC@5) with various $K_I$ and fixed $K_R = 128, K_L = 8, 16, 32, 64, 128$ respectively. We use text-embedding-3-large as the sentence encoder and set the candidate triples to $10^4$.

|             | $k_R = 8$ | $k_R = 16$ | $k_R = 32$ | $k_R = 64$ | $k_R = 128$ |
|-------------|-----------|------------|------------|------------|-------------|
| $k_I = 8$   | 2.38      | 2.38       | 2.38       | 11.9       | 11.9        |
| $k_I = 16$  | 0.0       | 2.38       | 2.38       | 19.2       | 19.2        |
| $k_I = 32$  | 1.92      | 3.85       | 3.85       | 34.6       | 34.6        |
| $k_I = 64$  | 7.14      | 7.69       | 9.62       | 50.0       | 50.0        |
| $k_I = 128$ | 57.6      | 11.5       | 7.69       | 61.5       | 61.5        |

Table 8: ACC@5 of inter-layer knowledge grounding accuracy on ATLAS-CC-QKV varying $k_R$ with fixed $k_I = 8, 16, 32, 64, 128$ respectively.

|             | $k_I = 8$ | $k_I = 16$ | $k_I = 32$ | $k_I = 64$ | $k_I = 128$ |
|-------------|-----------|------------|------------|------------|-------------|
| $k_L = 8$   | 9.62      | 19.2       | 28.8       | 34.6       | 36.5        |
| $k_L = 16$  | 14.3      | 21.2       | 35.7       | 50.0       | 48.1        |
| $k_L = 32$  | 11.5      | 19.2       | 36.5       | 46.2       | 61.5        |
| $k_L = 64$  | 11.5      | 19.2       | 34.6       | 50.0       | 59.6        |
| $k_L = 128$ | 11.5      | 19.2       | 34.6       | 51.9       | 63.5        |

Table 9: ACC@5 of root-layer knowledge grounding accuracy on ATLAS-CC-QKV varying $k_I$ with fixed $k_R = 128$ and $k_L = 8, 16, 32, 64, 128$ respectively.

As shown in Table 8 and Table 9, the results suggest that as we appropriately increase the upper-layer top-k values, the lower-layer accuracy will also continuously increase in most cases. But if too many keys are selected at the upper-layer, the accuracy at the lower-layer might also drop a little bit (e.g., $k_I = 128$ and $k_L = 16$). This observation also aligns with the conclusion described in Appendix B.2, which is mainly because the noise candidate keys selected at upper-layers would influence the selection accuracy of the lower-layers in HiKVP.

### B.5 BENIFITS OF KNOWLEDGE COVERAGE

In this section, we compare the benefits from knowledge coverage and the drawbacks from the noise introduced by larger KGs. To validate the benefits of knowledge coverage, we compared the performance of AtlasKV on a 100-triple sub-KG with a 10000-triple sub-KG from ATLAS-CC-QKV

| KGs | ATLAS-CC-QKV GPTScore | ATLAS-Pes2o-QKV GPTScore |
|---|---|---|
| $10^2$ Triples (w/o answers) | 10.8 | 13.6 |
| $10^4$ Triples (w/ answers) | **46.4** | **50.7** |

Table 10: GPTScore comparison between AtlasKV on different scaled KGs with and without answers.

and ATLAS-Pes2o-QKV, respectively, on a set of questions where the answers only exist in the larger sub-KG.

As shown in Table 10, the GPTScore of the responses generated with the larger KG is much higher than that of the responses generated with the smaller KG, which demonstrates that scaling up the KGs is valuable for knowledge augmented LLMs. And the benefit of knowledge coverage is much higher than the drawbacks from the noise introduced by the large scale KGs.

## C   DERIVATION OF THE ATTENTION MECHANISM IN ATLASKV

Here we give the details of how the attention mechanism in AtlasKV is derived from the standard rectangular attention in KBLaM.

*Proof.* First, the rectangular attention at the $l$-th attention layer and $n$-th token in KBLaM can be simply rewritten as:

$$\tilde{\mathbf{y}}_n^{(l)} = \frac{\sum_{i=1}^{M} \exp(\langle \tilde{\mathbf{q}}_n^{(l)}, \tilde{\mathbf{k}}^{(l)i}\rangle/\sqrt{D})\tilde{\mathbf{v}}^{(l)i}}{\sum_{i=1}^{M} \exp(\langle \tilde{\mathbf{q}}_n^{(l)}, \tilde{\mathbf{k}}^{(l)i}\rangle/\sqrt{D}) + \sum_{i=1}^{n} \exp(\langle \mathbf{q}_n^{(l)}, \mathbf{k}^{(l)i}\rangle/\sqrt{D})} + \tag{12}$$

$$\frac{\sum_{i=1}^{n} \exp(\langle \mathbf{q}_n^{(l)}, \mathbf{k}^{(l)i}\rangle/\sqrt{D})\mathbf{v}^{(l)i}}{\sum_{i=1}^{M} \exp(\langle \tilde{\mathbf{q}}_n^{(l)}, \tilde{\mathbf{k}}^{(l)i}\rangle/\sqrt{D}) + \sum_{i=1}^{n} \exp(\langle \mathbf{q}_n^{(l)}, \mathbf{k}^{(l)i}\rangle/\sqrt{D})}. \tag{13}$$

Then we replace the original formulation with the terms $\text{logits}_{kb}$ and $\text{logits}_{seq}$ as follows:

$$\tilde{\mathbf{y}}_n^{(l)} = \frac{\sum_{i=1}^{M} \exp(\text{logits}_{kb}^i)\tilde{\mathbf{v}}^{(l)i}}{\sum_{i=1}^{M} \exp(\text{logits}_{kb}^i) + \sum_{i=1}^{n} \exp(\text{logits}_{seq}^i)} + \tag{14}$$

$$\frac{\sum_{i=1}^{n} \exp(\text{logits}_{seq}^i)\mathbf{v}^{(l)i}}{\sum_{i=1}^{M} \exp(\text{logits}_{kb}^i) + \sum_{i=1}^{n} \exp(\text{logits}_{seq}^i)}, \tag{15}$$

where $\text{logits}_{kb}^i = \langle \tilde{\mathbf{q}}_n^{(l)}, \tilde{\mathbf{k}}^{(l)i}\rangle/\sqrt{D}$ and $\text{logits}_{seq}^i = \langle \mathbf{q}_n^{(l)}, \mathbf{k}^{(l)i}\rangle/\sqrt{D}$. Then we can calculate the softmax of the KB and sequence parts separately as follows:

$$\tilde{\mathbf{y}}_n^{(l)} = \underbrace{\frac{\sum_{i=1}^{M} \exp(\text{logits}_{kb}^i)}{\sum_{i=1}^{M} \exp(\text{logits}_{kb}^i) + \sum_{i=1}^{n} \exp(\text{logits}_{seq}^i)}}_{\lambda_{kb}} \cdot \underbrace{\frac{\sum_{i=1}^{M} \exp(\text{logits}_{kb}^i)\tilde{\mathbf{v}}^{(l)i}}{\sum_{i=1}^{M} \exp(\text{logits}_{kb}^i)}}_{\text{Softmax}(\text{logits}_{kb}^i)\tilde{\mathbf{v}}^{(l)i}} + \tag{16}$$

$$\underbrace{\frac{\sum_{i=1}^{n} \exp(\text{logits}_{seq}^i)}{\sum_{i=1}^{M} \exp(\text{logits}_{kb}^i) + \sum_{i=1}^{n} \exp(\text{logits}_{seq}^i)}}_{\lambda_{seq}} \cdot \underbrace{\frac{\sum_{i=1}^{n} \exp(\text{logits}_{seq}^i)\mathbf{v}^{(l)i}}{\sum_{i=1}^{n} \exp(\text{logits}_{seq}^i)}}_{\text{Softmax}(\text{logits}_{seq}^i)\mathbf{v}^{(l)i}}, \tag{17}$$

where $\lambda_{kb}$ and $\lambda_{seq}$ are the weights of the two parts. And in this way, the attention computation of the KB and sequence parts can be separated so that we can improve the scalability as well as efficiency of the KB part individually in a more intuitive way.

In our implementation, we replace the attention of the KB part with the KG part and replace $\lambda_{kb}$ with $\lambda_{kg}$ in a more scalable way as we described. □

# D    DERIVATION OF THE TIME AND MEMORY COMPLEXITY OF HIKVP

Here we proof the time and memory complexity of HiKVP are $\mathcal{O}\left((C_t \sqrt[3]{M} + N) \cdot N \cdot D\right)$ and $\mathcal{O}\left((C_m \sqrt[3]{M} + N) \cdot (N + D)\right)$, where $C_t = 1 + k_R + k_I$ and $C_m = \max(1, k_R, k_I)$ are constants that are much smaller than $M$.

*Proof.* **First, we analyze the time and memory complexity of HiKVP step by step.** For the process of calculating the softmax of attention scores with the root-layer keys at each step and selecet top-$k_R$ relevant root-layer keys with a heap of size $k_R$ to fetch the connected inter-layer keys we need in the next step, the time complexity is:

$$\mathcal{O}\left(\sqrt[3]{M}D + \sqrt[3]{M}\log(k_R)\right). \tag{18}$$

The memory complexity to store and calculate the attention scores of the root-layer keys of this process is (before offloading the root-layer keys to the CPU memory):

$$\mathcal{O}\left(\sqrt[3]{M}(N + D)\right). \tag{19}$$

Then we repeat that process with the selected inter-layer keys that are connected to the pruned root-layer keys and fetch the top-$k_I$ relevant inter-layer keys with a heap of size $k_I$ to obtain the connected leaf-layer keys we need in the next step, which has a time complexity of:

$$\mathcal{O}\left(k_R \sqrt[3]{M}D + k_R \sqrt[3]{M}\log(k_I)\right). \tag{20}$$

The memory complexity to store and calculate the attention scores of the selected inter-layer keys of this process is (after offloading the root-layer keys to the CPU memory and before offloading the selected inter-layer keys to the CPU memory):

$$\mathcal{O}\left(k_R \sqrt[3]{M}(N + D)\right). \tag{21}$$

Finally, we compute the softmax of the attention scores of the selected leaf-layer keys that are connected to the pruned inter-layer keys. Similarly, we also need to fetch the top-$k_L$ relevant leaf-layer keys with a heap of size $k_L$, which has a time complexity of:

$$\mathcal{O}\left(k_I \sqrt[3]{M}D + k_I \sqrt[3]{M}\log(k_L)\right). \tag{22}$$

The memory complexity to store the selected leaf-layer keys and pruned KG values, and to calculate their attention scores is (after offloading the selected inter-layer keys to CPU memory and before offloading the selected leaf-layer keys to GPU memory):

$$\mathcal{O}\left(k_I \sqrt[3]{M}(N + D)\right). \tag{23}$$

Note that $\overline{\text{logits}}_{kg}$ can be obtained by simplying selecting from $\overline{\text{logits}}_{kg_L}$ with the top-$k_L$ softmax scores indices. So this process would not take any additional time or memory complexity.

**Then we can synthesize the time and memory complexity of HiKVP at each step.** HiKVP has a total time complexity of:

$$\mathcal{O}\left((1 + k_R + k_I)D\sqrt[3]{M} + \left(\sqrt[3]{M}\log(k_R) + k_R\sqrt[3]{M}\log(k_I) + k_I\sqrt[3]{M}\log(k_L)\right)\right), \tag{24}$$

which can be simplified to:

$$\mathcal{O}\left((1 + k_R + k_I)D\sqrt[3]{M} + (\log(k_R) + k_R\log(k_I) + k_I\log(k_L))\sqrt[3]{M}\right). \tag{25}$$

And because usually $D \gg (\log(k_R) + k_R\log(k_I) + k_I\log(k_L))$, we can further simplify it to:

$$\mathcal{O}\left(C_t D\sqrt[3]{M}\right), \tag{26}$$

where $C_t = 1 + k_R + k_I$ is a constant. Then the total time complexity of both HiKVP part and the sequence part at all steps can be represented as:

$$\mathcal{O}\left((C_t\sqrt[3]{M} + N)ND\right), \tag{27}$$

Then for the total memory complexity of the both HiKVP and the sequence part at all steps, we have:

$$\mathcal{O}\left(\left(\max(1 + k_R + k_I)\sqrt[3]{M} + N\right)(N + D)\right), \tag{28}$$

which can be simplified to:

$$\mathcal{O}\left((C_m\sqrt[3]{M} + N)(N + D)\right), \tag{29}$$

where $C_m = \max(1, k_R, k_I)$ is a constant. □

## E   TRAINING DYNAMICS IN ATLASKV

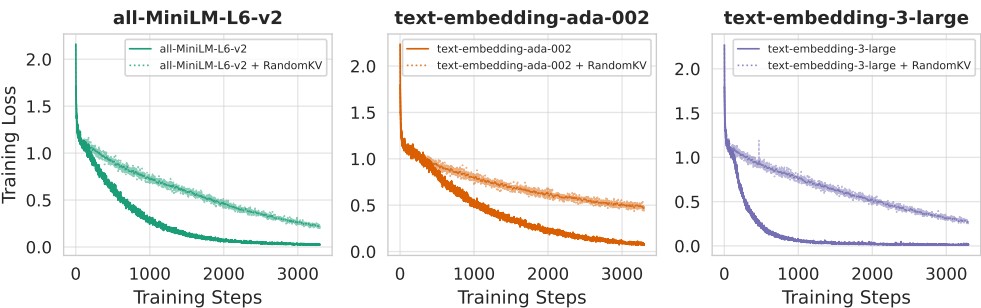

Figure 8: The training loss curves of AtlasKV with correct and random paired key-value embeddings (KGKVs) across three different sentence encoders.

We also observed dynamics in the training processes of AtlasKV, which can suggest *the model regularly start learning to retrieve relevant knowledge from the external KG triples, instead of brute force over-fitting, from a specific training step*. As shown in Figure 8, we trained AtlasKV on both correct and randomly paired KGKVs (denoted as "+ RandomKV") of ATLAS-Wiki-QKV dataset across three different sentence encoders, including all-MiniLM-L6-v2, text-embedding-ada-002, and text-embedding-3-large, respectively. We can find that before a specific training step, the training loss on these two variants of the dataset are almost the same. However, after that, the training loss on the correct variant of the dataset drops significantly while the training loss on the random variant of the dataset continues to decrease slowly. This suggests that AtlasKV starts to generalize by retrieving relevant knowledge from the external KG triples, instead of brute force over-fitting through neural parameters, from a specific training step. And this phenomenon can also support the experimental results that AtlasKV can achieve strong generalization abilities in OOD scenarios.

## F   WHERE IS THE TRADE-OFF?

### F.1   ACCURACY VS. SCALABILITY

The trade-off of AtlasKV is essentially between the performance and scalability. As expressed by Equation 27 and Equation 29, we can select different $k_R, k_I$ according to the specific needs to trade off the performance and scalability. For example, if we want to achieve higher performance, we can set a larger $k_R, k_I$. However, if we need higher scalability, we can set a smaller $k_R, k_I$ like our default settings in our previous experiments. Note that our experiments have suggested that AtlasKV can still achieve superior performance even with small $k_R, k_I$ while maintaining a good scalability. And if we do not prune any key at each layer, the scalability of AtlasKV will degenerate to that of the standard rectangular attention in KBLaM.

### F.2   INFERENCE LATENCY VS. SCALABILITY

From the inference latency perspective, there will be CPU-GPU I/O cost introduced in the HiKVP process. We conduct experiments to see how the hardware latency will change as the size of KGs scales up. As shown in Table 11, the results suggest that HiKVP will introduce some hardware latency. This is mainly because HiKVP need to upload or offload the keys between CPU and GPU memory during the hierarchical pruning process. So there is a trade-off between the inference latency and scalability. Some system scheduling algorithms might be needed to address this issue.

| Method | $10^1$ Triples Avg. Time (s) | $10^2$ Triples Avg. Time (s) | $10^3$ Triples Avg. Time (s) | $10^4$ Triples Avg. Time (s) |
|---|---|---|---|---|
| AtlasKV (128-64-16) | 4.10 | 4.02 | 4.44 | 6.21 |
| AtlasKV w/o HiKVP | 1.00 | 1.98 | 2.11 | 2.09 |
| Latency | 3.10 | 2.04 | 2.33 | 4.12 |

Table 11: Inference time comparisons of AtlasKV and AtlasKV w/o HiKVP on various KG sizes.

### F.3 ACCURACY VS. KG2KV COSTS

As discussed in Appendix B.2, if relation rewriting is not performed in either the training or testing data, there will be a slight performance drop in knowledge grounding accuracy. We do not need very large scale KGs to train AtlasKV, so the removal of relation rewriting in the training data is not necessary. However, as the scale of KGs in testing data increases, the relation rewriting process in KG2KV will introduce considerable token costs. So there is a trade-off between the accuracy and KG2KV costs when dealing with large scale KGs, which can be controlled by the choice of whether to perform relation rewriting process in KG2KV.

## G CASE STUDY

### G.1 DIFFERENCES BETWEEN SYNTHETIC KB AND KGKVS

In this section, we give some samples of the Q-K-V training data constructed by synthetic and KG2KV methods, respectively, to further demonstrate the differences between them. As shown in Table 12, we demonstrate the Q-K-V strings constructed by synthetic method and there are very limited and fixed enquiry attributes. However, as shown in Table 13, the Q-K-V strings constructed by KG2KV method have much more varied and flexible enquiry attributes, which are much more near to the real-world scenarios.

| *Q* | *K* | *V* |
|---|---|---|
| What is the *description* of Elara Moonshadow? | the *description* of Elara Moonshadow | The *description* of Elara Moonshadow is a skilled botanist with a passion for rare plants. |
| Describe the *description* of Thorne Blackwood? | the *description* of Thorne Blackwood | The *description* of Thorne Blackwood is a renowned chef known for his innovative culinary techniques. |
| Provide details on the *objectives* of Zara Nightingale? | the *objectives* of Zara Nightingale | The *objectives* of Zara Nightingale is to perform in prestigious concert halls worldwide. |
| Can you let me know the *purpose* of Lyra Starfire? | the *purpose* of Lyra Starfire | The *purpose* of Lyra Starfire is to preserve marine biodiversity. |
| Can you explain the *description* of Jaxon Wildheart? | the *description* of Jaxon Wildheart | The *description* of Jaxon Wildheart is a tech entrepreneur with a knack for innovative solutions. |
| What insights can you provide about the *objectives* of Kaelith Silverwind? | the *objectives* of Kaelith Silverwind | The *objectives* of Kaelith Silverwind is to document endangered animals. |

Table 12: Samples from Synthetic dataset. The enquiry attributes have been marked in *italics*.

### G.2 SAMPLE Q&A

As shown in Figure 9, we provide Q&A samples of AtlasKV, KBLaM and ICL with 100 triples in the ATLAS-CC-QKV dataset as candidates. We can tell that AtlasKV can generate a very relevant answer, which is almost close to the ICL's answer. However, KBLaM can not generate a relevant answer and even cannot provide any usefull information. This is mainly because KBLaM is limited by the fully synthetic training data and cannot be generalized to this unseen enquiry attribute. AtlasKV can achieve a higher relevant answer because of a higher diversity of the training data constructed by our KG2KV method.

| *Q* | *K* | *V* |
|---|---|---|
| What is the *explanation* of Postsocialist scholars? | the *explanation* of Postsocialist scholars | The *explanation* of Postsocialist scholars is the developments as a backlash against the 'feminizing' nature of the socialist state. |
| Can you explain the *cause* of World records? | the *cause* of World records | The *cause* of World records is World records in Paralympic powerlifting are ratified by the International Paralympic Committee. |
| Can you elaborate on the *rank* of Ramble On? | the *rank* of Ramble On | The *rank* of Ramble On is number 5 on the list of the 40 greatest Led Zeppelin songs. |
| How would you describe the *favorite* of Dick the Mockingbird? | the *favorite* of Dick the Mockingbird | The *favorite* of Dick the Mockingbird is among at least four mockingbirds the president had while in office. |
| Can you inform me about the *publication* of Hensley? | the *publication* of Hensley | The *publication* of Hensley is Fifty Miles from Tomorrow , a memoir of Alaska and the real people. |
| Tell me about the *threat* of African coral reefs? | the *threat* of African coral reefs | The *threat* of African coral reefs is industrial run-offs and pollutants, untreated sewage and the increasing sediment flows in rivers. |

Table 13: Samples from ATLAS-Wiki-QKV dataset. The enquiry attributes have been marked in *italics*.

---

**Sample Outputs.**

**Relevant Triple**: (*MOROCCO*; *consider*; *synthetic biology should be considered as a new and emerging issue*)
**Q**: *Can you elaborate on the opinion of MOROCCO?*
**K**: *the opinion of MOROCCO*
**V**: *The opinion of MOROCCO is synthetic biology should be considered as a new and emerging issue.*

- - - - - - - - - - - - - - - - - - - - - - - - - - - - - - - - - - - - - - - - - - - - - - - - - - -

**AtlasKV Output**:
The opinion of MOROCCO is issue of synthetic biology should be considered as a new frontier.

**KBLaM Output**:
I'm not sure what you mean. Can you provide more context?

**ICL Output**:
The opinion of MOROCCO is synthetic biology should be considered as a new and emerging issue.

Figure 9: A sample Q&A of AtlasKV, KBLaM, and ICL.

# H   PROMPT TEMPLATE

In this section, we give the prompt template we use to conduct the evaluations and KG2KV process. As shown in Figure 10, we use LLMs to generate the KGKVs from the text. And in this prompt template, we only need to provide the relation phrase and the missing entity position to generate the natural noun, which is token efficient. And this process usually do not need very powerful LLMs, which are cheaper. As shown in Figure 11, we use LLMs to score the relevance between the generated text and the ground truth answer. This process usually need powerful LLMs like GPT-4o, because it needs to evaluate the results with high quality.

---

**Prompt Template for KG2KV.**

**System Message**:

**Task:** Convert relation phrase to natural noun based on missing entity position.
**Rules:**
- **Missing head**: Passive relations → agent nouns ("govern" → "governor", "is participated by" → "participation")
- **Missing tail**: Active relations → object nouns ("produces" → "product", "achieves" → "achievement")

**Output:** Natural noun only.

**Examples:**
- ("is participated by", "head") → "participation"
- ("is participated by", "tail") → "participant"
- ("produces", "head") → "producer"
- ("produces", "tail") → "product"

- - - - - - - - - - - - - - - - - - - - - - - - - - - - - - - - - - - - - - - - - - - - - -

**User Message**:

relation: {relation}, missing: {missing}

---

Figure 10: The prompt template to rewrite the relation phrase to natural noun based on missing entity position in KG2KV process.

## I  THE USAGE OF LLMS

In this work, we use the LLM Claude-4-sonnet to polish statements and to check grammars in our paper. We also use that to help with our software developing, such as finding some issues in the codes and giving some advice to make the code structure better.

## J  LIMITATIONS AND FUTURE WORK

Although AtlasKV introduces a parametric method for integrating billion-scale KGs into LLMs without external retrievers, long contexts, or retraining, it still faces some limitations and challenges: (1) Loss of Graph Structure: The KG2KV method used in AtlasKV treats each triple in the KGs as an independent fact, which "flattens" the KGs. Even though AtlasKV can retrieve various knowledge at each generation or reasoning step, which can to some extent make up for the drawbacks of "flatten" KGs, this design will still block the LLMs' multi-hop reasoning capabilities on the KGs. (2) The precision drops in HiKVP: There is a trade-off between the accuracy and scalability controlled by various top-k values. In the future, different attention heads could be trained at different layers of HiKVP to further improve both fuzzy and precise retrieval capabilities. (3) Systematic cost in HiKVP: There will be CPU-GPU I/O cost during the HiKVP process. Some system scheduling algorithms could be introduced to further reduce this systematic cost.

---

Prompt Template for Relevance Scoring.

**System Message**:
You are an AI system that evaluates the quality of generated text. You will be given a text and a ground truth answer, your goals is to return a score between 0 and 1.

- - - - - - - - - - - - - - - - - - - - - - - - - - - - - - - - - - - - - - - - - - - - - - - - -

**User Message**:
Given a text and a ground truth answer, evaluate the quality of the text. Return a score of 1 if the text is exactly the same as the ground truth answer. Return a score of 0 if the text is completely wrong. Return a score between 0 and 1 if the text is partially correct. A more correct text should have a higher score. Do NOT generate anything else.
Example:
Model output: "The sky is blue."
True answer: "The sky is blue."
Score: **1.0**

Example 2:
Model output: "The color of Alexandria is blue."
True answer: "The color of Alexandria is green."
Score: **0.0**

Example 3:
Model output: "The purpose of Alexandria is to extract knowledge."
True answer: "The color of Alexandria is to discover and organize knowledge into a structured form."
Score: **0.9**

**Important**: Only generate a number.

---

Figure 11: The prompt template for the GPT-4o to score the relevance between the generated text and the ground truth answer.

