# OpenReview forum: "AtlasKV: Augmenting LLMs with Billion-Scale Knowledge Graphs in 20GB VRAM"
_ICLR.cc/2026/Conference — ICLR 2026 Poster_

### Official Review · Reviewer_yJsc · 2025-10-28

**Soundness:** 2
**Presentation:** 3
**Contribution:** 3
**Rating:** 4
**Confidence:** 4

**Summary:**

This paper proposes AtlasKV, a parametric knowledge augmentation method designed to integrate billion-scale Knowledge Graphs (KGs) into Large Language Models (LLMs) with extremely low VRAM cost (under 20GB). The core contributions are twofold: 1) KG2KV, a method for transforming KG triples into high-quality Query-Key-Value (Q-K-V) data, which aims to improve generalization by increasing query diversity compared to prior work. 2) HiKVP (Hierarchical Key-Value Pruning), a novel inference-time algorithm that organizes keys into a 3-layer hierarchy, enabling sub-linear ($\mathcal{O}(\sqrt[3]{M})$) time and memory complexity. This allows the model to scale to 1B triples while avoiding the linear cost growth seen in methods like KBLaM.

**Strengths:**

- Excellent Scalability: The primary strength of this paper is the HiKVP algorithm. It provides a credible path to scaling parametric knowledge augmentation to the billion-fact scale, a task that seems infeasible for prior methods like KBLaM. The experimental results in Figure 4, showing the ability to handle 1B triples with <20GB VRAM while KBLaM fails at 100K triples, are very impressive.

- Strong Generalization and Robustness: The KG2KV method effectively addresses the limited query diversity of template-based synthetic data used in prior work. This is validated by the strong results in Table 3, where AtlasKV (even with pruning) massively outperforms KBLaM on OOD datasets, especially complex ones like ATLAS-Pes20-QKV (e.g., 52.7% vs 5.5% on 10³ triples).

- Addresses a Critical Problem: Efficiently and scalably augmenting LLMs with vast, external factual knowledge is a fundamental goal in AI. This paper tackles this problem head-on with a novel parametric approach that avoids the high inference latency of RAG.

**Weaknesses:**

- Major Flaw in Experimental Validation: The paper's core premise is that scaling up KGs to 1B triples is valuable. However, the experiments fail to demonstrate this value. Table 3, the primary knowledge grounding experiment, only shows that as the number of "candidate triples" (i.e., noise) increases, performance decreases. This experiment proves "robustness to noise" but critically fails to prove the benefit of "knowledge coverage".
The paper is missing the most crucial experiment: a comparison between a small KG and the large KG on a set of questions where the answers only exist in the large KG. Without this, the motivation for the entire paper—scaling up—is left as an unsubstantiated assumption. As it stands, the experiments show that scaling up (the candidate pool) only hurts performance.

- Unaccounted LLM Cost & Unfair Comparison: The KG2KV method (Section 4.1, Appendix H) relies on external, powerful LLMs (GPT-4o-mini, and the authors state in Appendix I that Claude-4-sonnet was used for development) to transform KG relations into natural noun phrases. This introduces a significant, one-off (but potentially massive) computational and/or API cost for preprocessing the 1B+ triples. This cost is completely ignored in the paper's efficiency analysis. This makes the comparison to methods like KBLaM, which uses simple synthetic templates, an unfair, apples-to-oranges comparison. The scalability of AtlasKV is thus partially achieved by offloading significant complexity to an external, unmeasured LLM.

- Loss of Graph Structure: The KG2KV method inherently "flattens" the knowledge graph, treating each triple as an independent fact. This sacrifices all topological/structural information of the KG, limiting the method to one-hop retrieval and precluding multi-hop reasoning. This is a significant trade-off that should be more explicitly acknowledged and discussed as a limitation.

**Questions:**

- Could the authors provide an experiment that demonstrates the value of "knowledge coverage"? For example, by comparing a model trained on a 1M-triple KG vs. a 100M-triple KG on a set of questions where the answers only exist in the larger KG? This seems essential to justify the paper's core motivation.

- Can the authors quantify the computational cost (e.g., in GPU hours or total API costs) of the KG2KV preprocessing step for the 1B-triple ATLAS-Wiki dataset? How does this one-off cost compare to the training/inference costs, and how does it affect the overall efficiency claim?

- The performance of AtlasKV with HiKVP is notably lower than AtlasKV without HiKVP (e.g., Table 3, ATLAS-CC-QA, 10³ triples: 74.5% vs 96.4%). This suggests the hierarchical pruning is quite lossy. Have the authors explored strategies to mitigate this precision drop (e.g., different top-k values, as hinted at in Appendix B.2), or is this considered an acceptable trade-off for the achieved scalability?

---

> ### Author Response · Authors · 2025-11-19
> **Response to Reviewer yJsc (1/2)**
>
> We appreciate Reviewer yJsc for the detailed and constructive comments. We would like to address the concerns raised as follows:
>
> > W1 & Q1. Lack the experiment that demonstrates the value of "knowledge coverage".
>
> Thank you for your insightful feedback regarding the comparison between the benefits from "knowledge coverage" and the drawbacks from the noise introduced by larger KGs. To validate the value of "knowledge coverage", we compared the performance of AtlasKV on a 100-triple sub-KG from ATLAS-CC-QKV with a 10000-triple sub-KG from ATLAS-CC-QKV on a set of questions where the answers only exist in the larger sub-KG.
>
> |KGs|GPTScore|
> |-|-|
> |$10^2$ Triples (w/o answers)|10.8|
> |$10^4$ Triples (w/ answers)|46.4|
>
> As shown in the above table, the GPTScore of the responses generated with the larger KG is much higher than that of the responses generated with the smaller KG, which demonstrates that **scaling up the KGs is valuable for knowledge augmented LLMs**. And **the benefit of "knowledge coverage" is much higher than the drawbacks from the noise introduced by the large scale KGs**.
>
> We have the above experiments into Appendix B.5 of our revision.
>
> > W2 & Q2. Unaccounted LLM Cost & Unfair Comparison between KG2KV and the Synthetic method.
>
> Thank you for your valuable suggestions! We would like to clarify that from two perspectives: (1) **Compared with the fully Synthetic method, KG2KV is not only more effective but also cheaper**. (2) **Even without relation rewriting, which is the only process that consumes tokens in KG2KV, AtlasKV can also perform well**.
>
> (1) As described in Section 4.1 and shown in Table 1, compared with the fully Synthetic method, **the average token cost of processing each triple is more than 50% lower in KG2KV**, which is 165.7. Because the Synthetic method in KBLaM utilizes GPT-4 to generate both named entities and their attribute descriptions. This is a much harder task that needs predefined schemas as a longer context than the relation rewriting task in KG2KV. The only strings we need input into the LLM in KG2KV is the masked position and the relation. So **GPT-4o-mini is enough for KG2KV, which is much cheaper than GPT-4 used in KBLaM**.
>
> (2) Regarding the considerable token cost when applying KG2KV to billion-scale KGs, we conducted experiments with a naive version of AtlaskV, which is denoted as $\text{AtlasKV-Ntest}$. In the naive version of AtlasKV, we do not rewrite KG relations into natural noun phrases with LLMs. We directly use the combination of original relations and unmasked entities as the keys. We still used the model trained on ATLAS-Wiki-QKV, and tested the performance on ATLAS-Pes2o-QKV without relation rewriting on that.
>
> |Method|$10^1$ Triples||$10^2$ Triples||$10^3$ Triples||$10^4$ Triples||
> |-|-|-|-|-|-|-|-|-|
> ||ACC@1|ACC@5|ACC@1|ACC@5|ACC@1|ACC@5|ACC@1|ACC@5|
> |KBLaM|50.0|$\underline{80.0}$|25.5|52.7|3.6|14.5|0.0|5.5|
> |AtlasKV w/o HiKVP|$\textbf{100.0}$|$\textbf{100.0}$|$\textbf{92.7}$|$\textbf{100.0}$|$\textbf{72.7}$|$\textbf{90.9}$|$\textbf{47.3}$|$\textbf{67.2}$|
> |AtlasKV-Ntest w/o HiKVP|$\textbf{100.0}$|$\textbf{100.0}$|76.9|$\underline{98.0}$|$\underline{55.8}$|$\underline{78.8}$|$\underline{26.9}$|46.2|
> |AtlasKV|$\underline{90.0}$|$\textbf{100.0}$|$\underline{87.3}$|92.7|52.7|70.9|16.4|$\underline{49.0}$|
> |AtlasKV-Ntest|$\underline{90.0}$|$\textbf{100.0}$|86.5|90.4|46.2|61.5|23.1|36.5|
>
> As shown in the above results, **without the usage of LLMs (relation rewriting) in KG2KV, the naive version of AtlaskV can still perform well**. The performance drops are acceptable compared with the full model of AtlasKV. Although more accurate key strings are beneficial, simple combination of relation and unmasked entity strings can also be very helpful for distinguishing them. So people can either choose AtlasKV or AtlasKV-Ntest according to their tradeoff among accuracy, token consumption and KG scale.
>
> We have added the above experiments into Appendix B.2 of our revision. We also added the discussions of the trade-off between accuracy and token costs in KG2KV, as revealed in Appendix F.3 of our revision.
>
> > Q3. Loss of Graph Structure.
>
> Thank you for your insightful comments! We would like to clarify that as verified in KBLaM, **this knowledge augmentation paradigm can also ground multiple triples**. Furthermore, the knowledge augmentation paradigm used in AtlasKV can retrieve and inject external knowledge at each prefilling and decoding step. So actually **AtlasKV can retrieve various external knowledge according to the requirements at each generation or reasoning step, which can to some extent make up for the drawbacks of "flatten" KGs from KG2KV**. We will consider improving the multi-hop reasoniong capabilities of AtlasKV with  as a very important future direction.
>
> We have also added this discussion into the limitation and future work section Appendix J of our revision.

---

> ### Author Response · Authors · 2025-11-19
> **Response to Reviewer yJsc (2/2)**
>
> Follows up to Response to Reviewer yJsc (1/2).
>
> > W3. Concerns about the precision drop in HiKVP.
>
> Thank you for your valuable feedback. We would like to clarify that  although there will be some performance drop in HiKVP for the AtlasKV's scalability, the performance drop can be relieved by increasing the $k_{R}, k_{I}$ and $k_{L}$. As verified in Appendix B.2 (Appendix B.4.1 in the revision) and discussed in Appendix F, as we appropriately increase these top-k values, the performance can be further improved. The boundary case is that if we do not prune any key at each layer of HiKVP, the scalability of AtlasKV will degenerate to that of the standard rectangular attention in KBLaM. So **there is a trade-off between the accuracy and scalability that can be easily controlled through selecting different top-k values according to the users' requirements**. We also believe how to further improve the AtlasKV's pruning accuracy is a valuable research problem for the future work to address.
>
> We have added discussions of the trade-off between accuracy and scalability of HiKVP into Appendix F.1 of our revision.

---

> ### Author Response · Authors · 2025-11-27
> **Your feedback to our rebuttal would be greatly appreciated**
>
> Dear Reviewer yJsc,
>
> Thank you again for the constructive comments you gave us in your review. As the rebuttal phase will end on Dec 3, we would greatly appreciate it if you could also take some time to check if our rebuttal has addressed your concerns, and please let us know if you would like us to provide any further clarification about the concerns you have.
>
> Best,
>
> Authors

---

### Official Review · Reviewer_K64s · 2025-10-29

**Soundness:** 2
**Presentation:** 3
**Contribution:** 2
**Rating:** 4
**Confidence:** 3

**Summary:**

This paper proposes a relatively systematic optimization framework to address the high computational complexity of knowledge-augmented large language models (KG-augmented LLMs). It provides both theoretical and experimental analyses of the model’s time and space complexity as well as GPU memory consumption, showing a certain level of exploratory research value. The designs of the hierarchical knowledge base and Rectangular Attention modules are instructive. In addition, the authors present relatively complete complexity derivations and visualization results, which enhance the technical rigor of the work.

**Strengths:**

1. **Novel and principled framework:**

The transformation from KG triples to Q-K-V structures is conceptually elegant and theoretically aligned with transformer attention, effectively bridging symbolic and neural representations.

2. **Scalable design and sub-linear efficiency:**

The hierarchical pruning mechanism (HiKVP) achieves sub-linear time and memory complexity, making it feasible to integrate billion-scale KGs in limited VRAM environments.

3. **Strong empirical results:**

Extensive experiments demonstrate clear improvements in both knowledge grounding and generalization, with substantial reductions in memory cost compared to KBLaM and RAG.

**Weaknesses:**

1. The methodology section is difficult to follow. The main issue is that many symbols are not clearly defined. For example, the meaning of $S$ on Line 253 and the correspondence of variables in Eq. (3) and Eq. (13). In addition, the paper does not state the model’s optimization objective, which further increases the burden on readers trying to understand the work.

2. The paper’s primary contribution appears to be complexity optimization for KG-augmented LLMs. However, relying solely on time/space complexity analyses and VRAM accounting is not sufficient to fully assess this advantage. For instance, when performing top-k pruning over a hierarchical KB, switching among three levels of keys can induce frequent CPU–GPU I/O, which is a non-negligible hardware latency during real-world training/inference. I recommend discussing this overhead explicitly.

3. Missing parameter sensitivity analysis. To my knowledge, Rectangular Attention in KBLaM is not invoked at every LLM layer; it is typically applied every few layers. I could not find this setting in the main text (it might be in the appendix, which I may have overlooked). Moreover, the paper does not provide any rationale or analysis for such a setting.

4. Why three hierarchy levels? The choice of a 3-level hierarchical key-value cache is insufficiently justified. Please add more detailed theoretical or empirical support for this design choice.

5. Inter-level correlation in top-k selection is not clear. Are the top-k selections across different hierarchy levels correlated? Figure 6 only shows the variation of a single variable. Given that top-k pruning at an upper level can influence the quality of pruning at the next level, an experimental analysis of this inter-level dependency seems necessary.

6. I am curious about the appendix proof of equivalence for Rectangular Attention. It appears to be largely a distributive rearrangement of matrix operators that, in practice, does not change the computed result. Please provide a deeper explanation of its effectiveness. Alternatively, clarify whether this operator manipulation merely serves as an algebraic device to facilitate subsequent complexity analysis for hierarchical key-value computation.

**Questions:**

Please See weakness

---

> ### Author Response · Authors · 2025-11-19
> **Response to Reviewer K64s (1/2)**
>
> We appreciate Reviewer K64s for highlighting our work’s novelty, scalability and strong empirical results. We would like to address your concerns below:
>
> > W1: The methodology section is difficult to follow.
>
> We are sorry for your confusion about our defined symbols and variables in the equations. We would like to further illustrate the symbols and variables you mentioned.
>
> $S = \left \lceil \sqrt[3]{M}\right \rceil$ is the size of clusters at each layer in HiKVP, where $M$ is the number of triples of the KG and has been defined in Section 3.2. In Eq. (3) and Eq. (13), $\tilde{\textnormal{y}}^{(l)}_{n}$ represents the output of the $l$-th layer attention network and the definition of other variables are illustrated in Eq. (4), Eq. (5), Eq. (6) and Section 3.2. Compared with the standard attention network as demonstrated in Eq. (1), we separate it into an external KG part and a sequence part and apply HiKVP algorithm to the KG part attention to make it scalable.
>
> For our model’s optimization objective, we perform partial-parameters tuning of the LLM on the prediction tokens, using its original auto-regressive training objective. Mathematically, the training objective is to maximize the probability of generating target answers by $p(v | \mathcal{M}, q) =  \prod_{i=1}^{L} p_{\theta}(x_{i} | \mathcal{M}, q_{<i}, v_{<i})$, where $L$ is the totoal length of $q$ and $v$, $i$ means the $i$-th token in the combination of $q$ and $v$. $q_{<i}$ and $v_{<i}$ are the query and answer tokens in all turns before the current prediction token $x_i$. And $\theta$ represents the trainable parameters, which are only $\tilde{\mathbf{W}}_Q$, $\tilde{\mathbf{W}}_K$, and $\tilde{\mathbf{W}}_V$ in our work.
>
> We have added the illustrations into our revision in Section 4.2.
>
> > W2. Lack discussions about the hardware latency during real-world training/inference
>
> Thank you for pointing out this potential system problem in AtlasKV! This hardware latency will only exist in the inference time, because **during the training process, we do not need to train on such a large scale of KGs with HiKVP as described in Appendix A.1**. For the hardware latency during the inference time, there will be CPU-GPU I/O cost during the HiKVP process. We conducted experiments to see how the hardware latency will change as the size of KGs scales up. We compared the inference time between AtlasKV and AtlasKV w/o HiKVP.
>
> |Method|$10^1$ Triples|$10^2$ Triples|$10^3$ Triples|$10^4$ Triples|
> |-|-|-|-|-|
> ||Avg. Time (s)|Avg. Time (s)|Avg. Time (s)|Avg. Time (s)|
> |AtlasKV |4.10|4.02|4.44|6.21|
> |AtlasKV w/o HiKVP|1.00|1.98|2.11|2.09|
> |Latency|3.10|2.04|2.33|4.12|
>
> Even though the HiKVP algorithm will introduce some hardware latency, we believe it can be addressed through some system scheduling algorithm. We will consider this as a future direction for further improving the efficiency of AtlasKV from systematic perspectives.
>
> We have added the above experiments into Appendix F.2 of our revision. We also included that into the limitations in Appendix J of our revision.
>
> > W3. Lack parameter sensitivity analysis.
>
> Thank you for your valuable suggestions! In the comparison experiments of our work, we use the same knowledge injection frequency settings in AtlasKV and KBLaM, in which we set both of their frequency as 3. Because as verified in KBLaM, it's a compromise that doesn't consume too much memory or significantly impact performance.
>
> We also did experiments to analyze the parameter sensitivity in AtlasKV. Following the  experiments in KBLaM, we set the knowledge injection frequency $K$ as 1, 3, 10 and then compare the knowledge grounding accuracy on the ATLAS-Pes2o-QKV dataset.
>
> |Method|$10^1$ Triples||$10^2$ Triples||$10^3$ Triples||$10^4$ Triples||
> |-|-|-|-|-|-|-|-|-|
> ||ACC@1|ACC@5|ACC@1|ACC@5|ACC@1|ACC@5|ACC@1|ACC@5|
> |KBLaM|50.0|80.0|25.5|52.7|3.6|14.5|0.0|5.5|
> |AtlasKV ($K=1$) w/o HiKVP|$\underline{90.0}$|$\textbf{100.0}$|82.7|$\underline{94.2}$|$\underline{55.8}$|$\underline{76.9}$|$\underline{33.5}$|$\underline{55.8}$|
> |AtlasKV ($K=3$) w/o HiKVP|$\textbf{100.0}$|$\textbf{100.0}$|$\textbf{92.7}$|$\textbf{100.0}$|$\textbf{72.7}$|$\textbf{90.9}$|$\textbf{47.3}$|$\textbf{67.2}$|
> |AtlasKV ($K=10$) w/o HiKVP|80.0|$\underline{90.0}$|75.0|88.5|40.4|55.8|32.7|36.5|
> |AtlasKV ($K=1$)|$\underline{90.0}$|$\textbf{100.0}$|76.9|92.3|32.7|55.8|11.5|21.2|
> |AtlasKV ($K=3$)|$\underline{90.0}$|$\textbf{100.0}$|$\underline{87.3}$|92.7|52.7|70.9|16.4|49.0|
> |AtlasKV ($K=10$)|80.0|$\underline{90.0}$|67.3|84.6|15.4|38.5|7.69|25.0|
>
> As shown in the above results, in AtlasKV, too frequent or infrequent knowledge injection will both lead to suboptimal performance. $K=3$ is the best choice in AtlasKV. This is mainly because too frequent knowledge injection will introduce noise in early attention layers and with lower frequency, the model will fail to ground accurate triples due to inadequate knowledge injection.
>
> We have added the above experiments into Appendix B.3 of our revision.

---

> ### Author Response · Authors · 2025-11-19
> **Response to Reviewer K64s (2/2)**
>
> Follows up to Response to Reviewer K64s (1/2).
>
> > W4. Why three hierarchy levels?
>
> Indeed, the hierarchy levels can be extended to more than 3-level depending on specific requirements. The choice of 3-level in this work is based on the observation that **3 is the minimal number of levels that captures all the essential definitions and functionalities of our HiKVP algorithm**. This design provides a general and extensible framework: when more levels are needed, additional intermediate layers can be easily inserted without altering the core structure or principles of the hierarchy.
>
> > W5. Inter-level correlation in top-k selection is not clear.
>
> We sincerely appreciate your insightful suggestion! To address your concerns regarding the inter-level correlation in top-k selection of HiKVP, we further compared the knowledge grounding accuracy at each layer with different top-k selections at a higher layer. To be specific, to find the correlations of top-k values between the root layer and inter layer, we remove the pruning process in the leaf layer and compare the knowledge grounding performance with fixed $K_I$ and various $K_{R}$. The same  experiments are also conducted with fixed $K_{R}, K_{L}$ and various $K_{I}$.
>
> ||$k_R=8$|$k_R=16$|$k_R=32$|$k_R=64$|$k_R=128$|
> |-|-|-|-|-|-|
> ||ACC@5|ACC@5|ACC@5|ACC@5|ACC@5|
> |$k_I = 64$|7.14|7.69|9.62|50.0|50.0|
>
> ||$k_I=8$|$k_I=16$|$k_I=32$|$k_I=64$|$k_I=128$|
> |-|-|-|-|-|-|
> ||ACC@5|ACC@5|ACC@5|ACC@5|ACC@5|
> |$k_L=16$|14.3|21.2|35.7|50.0|48.1|
>
> The above results suggest that **as we appropriately increase the upper-layer top-k values, the lower-layer accuracy will also continuously increase. But if too many keys are selected at the upper-layer, the accuracy at the lower-layer will drop a little bit**. This observation also aligns with the conclusion described in Appendix B.2 (Appendix B.4.1 in the revision), which is mainly because **the noise candidate keys selected at upper-layers would inﬂuence the selection accuracy of the lower-layers in HiKVP**. We have added more comprehensive visualization results, implementation details, and discussions of the above experiments in Appendix B.4.2 in our revision.
>
> > W6. Questions about the appendix proof of equivalence for Rectangular Attention.
>
> We apologize for any confusion caused by the brevity of the appendix. The rectangular-attention rewrite is indeed an exact algebraic equivalence. In this way, we can separate the original attention into sequence part:
>
> $\lambda_{seq} \cdot \text{Softmax}\left(\text{logits}_{seq}\right) \cdot \mathbf{v}^{(l)}$,
>
> and the KG part:
>
> $\lambda_{kg} \cdot \text{Softmax}\left(\text{logits}_{kg_L}\right) \cdot \tilde{\mathbf{v}}^{(l)}$.
>
> This separation does not alter the computed result but can expose the KG component to targeted pruning and reuse, which subsequent complexity analyses and our hierarchical optimization rely on.

---

> ### Author Response · Authors · 2025-11-27
> **Your feedback to our rebuttal would be greatly appreciated**
>
> Dear Reviewer K64s,
>
> Thank you again for the constructive comments you gave us in your review. As the rebuttal phase will end on Dec 3, we would greatly appreciate it if you could also take some time to check if our rebuttal has addressed your concerns, and please let us know if you would like us to provide any further clarification about the concerns you have.
>
> Best,
>
> Authors

---

### Official Review · Reviewer_yVFP · 2025-10-30

**Soundness:** 3
**Presentation:** 3
**Contribution:** 3
**Rating:** 8
**Confidence:** 4

**Summary:**

This paper proposes AtlasKV, a parametric method for integrating billion-scale KGs into LLMs without external retrievers, long contexts, or retraining.

It addresses limitations of non-parametric RAG (latency, retrieval bottlenecks) and traditional parametric methods (retraining costs, poor scalability)

Key innovations include KG2KV, which converts KG triples into query-key-value (Q-K-V) data aligned with LLM attention mechanisms, and HiKVP, a hierarchical pruning algorithm for sub-linear time/memory complexity during inference.

Experiments demonstrate superior empirical performance.

**Strengths:**

1. AtlasKV shifts from retrieval-heavy RAG to parametric KG injection, enabling training-free adaptation to new KGs. This idea is both novel and practical.

2. The method is well-motivated, with clear formalisms for KG2KV and HiKVP. Such formulation maintains generalization without fine-tuning.

3. Results show strong gains in knowledge-intensive tasks.

**Weaknesses:**

1. HiKVP's pruning is efficient, but lacks formal guarantees or error analysis.

2. Lack comparison with GraphRAG algorithms, such as HippoRAG and Raptor. I understand the focus of this work is on large KGs, where existing graphRAG algorithms can struggle to handle especially for indexing latency and token consumption. But comparing with them on small datasets can benefit the understanding the pros and cons of these methods.

**Questions:**

See above.

---

> ### Author Response · Authors · 2025-11-19
> **Response to Reviewer yVFP (1/1)**
>
> We sincerely thank Reviewer yVFP for the valuable and constructive comments. We would like to address the concerns individually below:
>
> > W1. HiKVP's pruning is efficient, but lacks formal guarantees or error analysis.
>
> Thank you for your feedbacks! We would like to clarify that we conduct error analysis in Appenix B.2, in which we investigate how different top-k settings at each layer
> of HiKVP inﬂuence the knowledge grounding accuracy. We found that the accurate retrieval ability of AtlasKV is stronger than the fuzzy retrieval ability of it. So **the error cases are mainly due to the weak fuzzy retrieval ability and the noise candidate keys selected in early attention layers would inﬂuence the retrieval accuracy of the later attention layers**. For formal guarantees in HiKVP, we have analyzed the time and memory complexity of HiKVP theoretically in Appendix D, which demonstrates that **the complexity of HiKVP increase sub-linearlly as the KGs scale up**. In this way, AtlasKV trades the sub-linear attention-based retrieval cost ($O(C_{t}\sqrt[3]M)$,$O(C_{m}\sqrt[3]M)$) in HiKVP for the ability to perform efficient inference independent of the scale of the retrieved relevant knowledge.
>
> > W2. Lack comparison with GraphRAG algorithms on small datasets.
>
> Thank you for your suggestions! We further compare the GPTScores of the answers generated by AtlasKV with those of GraphRAG[1], and Raptor[2] based on 100-Triple KGs. We use the same LLM backbone LLaMA3.1-8B-Instruct as the answer generator and use the same sentence encoder all-MiniLM-L6-v2.
>
> |Method|GPTScore|
> |-|-|
> |GraphRAG|72.6|
> |Raptor|82.1|
> |AtlasKV|67.8|
>
> As shown in the above results, **without the in-context prior, AtlasKV can still perform closely to the GraphRAG on small datasets**. Although AtlasKV performs obviously worse than Raptor, note that different from Raptor and GraphRAG, AtlasKV do not need any extra token consumption, which is cheaper and more efficient.
>
> We will also consider how to incorporate both non-parametric and parametric knowledge augmentation methods to further improve the accuracy of AtlaskV while maintaining efficiency and low token consumption as a future improving direction.
>
> [1] Edge D, Trinh H, Cheng N, et al. From local to global: A graph rag approach to query-focused summarization[J]. arXiv preprint arXiv:2404.16130, 2024.
>
> [2] Sarthi P, Abdullah S, Tuli A, et al. Raptor: Recursive abstractive processing for tree-organized retrieval[C]//The Twelfth International Conference on Learning Representations. 2024.
>
> Thanks again for your recognition of our work and your valuable feedbacks!

---

> > ### Comment · Reviewer_yVFP · 2025-11-27
> > **Response to Authors**
> >
> > Dear Authors,
> >
> > Thank you for the detailed rebuttal. However, you have not addressed the inclusion of HippoRAG/HippoRAG-2, which I suggested in my review comments. Could you please provide these results if time permits?
> >
> > Moreover, your rebuttal raises some concerns regarding the practicability of this work. Based on the results you provided, graph-based RAG methods appear to achieve higher performance rather than performing "closely" as claimed. In my assessment, HippoRAG should demonstrate greater efficiency than Raptor in indexing time while maintaining better accuracy. Additionally, recent Graph-RAG advancements such as $E^2GraphRAG$[1] and $LinearRAG$[2] show promising potential for scaling to large knowledge bases. These factors may challenge the practical advantages of your approach. Could you include a comparative analysis or detailed discussion referencing these methods? I believe this would help in understanding the motivation and contribution of your work.
> >
> > I will maintain my current positive score for now, though my final decision will depend on your response to these points.
> >
> > - [1] E^2GraphRAG: Streamlining Graph-based RAG for High Efficiency and Effectiveness
> > - [2] LinearRAG: Linear Graph Retrieval Augmented Generation on Large-scale Corpora

---

> > > ### Author Response · Authors · 2025-11-29
> > > **Comparisons with RAG Baselines**
> > >
> > > Dear Reviewer yVFP,
> > >
> > > Thank you so much for your followup feedbacks! We would like to demonstrate the superiority of AtlasKV with further comparisons of accuracy (GPTScore $\uparrow$) and efficiency (Avg. Inference Latency $\downarrow$ and Avg. Context Length $\downarrow$) against HippoRAG-2[1], HippoRAG[2], GraphRAG[3], and Raptor[4]. To fairly compare them, we use the same locally deployed LLM backbone `Llama-3.1-8B-Instruct` and the same locally deployed sentence encoder `all-MiniLM-L6-v2` as the retriever. We set the topk equally as 16 to make the context length comparable, which is the same as the leaf-layer topk value of AtlasKV. And we set the `max_new_tokens` of their generation process equally as 512 to make the inference latency comparable.
> > >
> > > |Method|GPTScore $\uparrow$|Avg. Inference Latency (s) $\downarrow$|Avg. Context Length $\downarrow$|
> > > |-|-|-|-|
> > > |HippoRAG-2|74.6|13.30|1173.8|
> > > |HippoRAG|72.1|12.28|1173.8|
> > > |GraphRAG|72.6|16.66|1642.3|
> > > |Raptor|$\textbf{82.1}$|15.53|1454.8|
> > > |AtlasKV|67.8|$\textbf{4.02}$|$\textbf{0}$|
> > >
> > > As shown in the above table, even though the GPTScore of AtlasKV is slightly lower (67.8 vs. 72.1, 72.6, and 74.6) than the ICL-based RAG methods with small dataset, **the average inference latency and context length of AtlasKV are significantly lower (4.02 vs. 12.28, 0 vs. 1173.8) than these baselines**. Because different from the knowledge augmentation paradigm of ICL-based RAG baselines, AtlasKV does not require any long-context prior to augment LLMs with the relevant external knowledge. Especially when the relevant external knowledge scales up, the context length and inference latency of RAG methods will become much higher and intolerable. So **taking both of accuracy and efficiency factors into consideration, AtlasKV  demonstrates strong superiority against ICL-based RAG methods.**
> > >
> > > For some recent graph-RAG advancements during the same period as AtlasKV for scaling to large knowledge bases, $E^2$GraphRAG[5] constructs bidirectional indexes between entities and chunks to enable fast lookup during both local and global retrieval. LinearRAG[6] constructs a relation-free hierarchical graph to let graph construction scale linearly with corpus size. Even though they have achieved some successes for indexing large scale external knowledge bases, **they still follow the ICL-based RAG paradigm, relying on external retrieval to augment LLMs with contextual relevant knowledge during inference, which will still face the inference efficiency challenge as we described in last paragraph**. However, **the long-context-prior-free and  external-retriever-free knowledge augmentation paradigm applied in AtlasKV demonstrates practical advantages of inference efficiency**.
> > >
> > > Additionally, as mentioned in the paper, note that **the graph construction process is not the focus of our work. And any existing constructed KGs can be applied in AtlasKV to augment LLMs in a scalable, effective, and general way**.
> > >
> > > We have also included the above discussions into our revision in Section 2. Thank you again for your constructive and valuable suggestions!
> > >
> > > [1] Gutiérrez, Bernal Jiménez, et al. "From rag to memory: Non-parametric continual learning for large language models." arXiv preprint arXiv:2502.14802 (2025).
> > >
> > > [2] Jimenez Gutierrez, Bernal, et al. "Hipporag: Neurobiologically inspired long-term memory for large language models." Advances in Neural Information Processing Systems 37 (2024): 59532-59569.
> > >
> > > [3] Edge D, Trinh H, Cheng N, et al. From local to global: A graph rag approach to query-focused summarization[J]. arXiv preprint arXiv:2404.16130, 2024.
> > >
> > > [4] Sarthi P, Abdullah S, Tuli A, et al. Raptor: Recursive abstractive processing for tree-organized retrieval[C]//The Twelfth International Conference on Learning Representations. 2024.
> > >
> > > [5] Zhao, Yibo, et al. "E^ 2GraphRAG: Streamlining Graph-based RAG for High Efficiency and Effectiveness." arXiv preprint arXiv:2505.24226 (2025).
> > >
> > > [6] Zhuang, Luyao, et al. "LinearRAG: Linear Graph Retrieval Augmented Generation on Large-scale Corpora." arXiv preprint arXiv:2510.10114 (2025).

---

> ### Author Response · Authors · 2025-11-27
> **Your feedback to our rebuttal would be greatly appreciated**
>
> Dear Reviewer yVFP,
>
> Thank you again for the constructive comments you gave us in your review. As the rebuttal phase will end on Dec 3, we would greatly appreciate it if you could also take some time to check if our rebuttal has addressed your concerns, and please let us know if you would like us to provide any further clarification about the concerns you have.
>
> Best,
>
> Authors

---

### Official Review · Reviewer_bR9N · 2025-11-03

**Soundness:** 3
**Presentation:** 3
**Contribution:** 3
**Rating:** 8
**Confidence:** 3

**Summary:**

This paper introduces AtlasKV, a parametric knowledge integration framework for large language models (LLMs) that enables augmentation with billion-scale knowledge graphs (KGs) under modest GPU memory budgets (<20GB VRAM). The method converts KG triples into query–key–value (QKV) training data aligned with LLM attention structures. It also employs a hierarchical key–value pruning algorithm that reduces computational and memory overhead while preserving grounding accuracy. Extensive experiments on large-scale KG datasets (ATLAS family, Enron, etc.) demonstrate that AtlasKV outperforms both non-parametric RAG methods and parametric baselines (e.g., KBLaM) in terms of scalability, knowledge grounding, and generalization to out-of-distribution (OOD) queries.

**Strengths:**

- The combination of KG2KV and HiKVP is elegant, bridging symbolic KG structure with LLM attention in a natural way.

- The proposed method demonstrates integration of 1B triples with <20GB VRAM, a significant improvement over KBLaM (>40GB for 100K triples).

- Experiments across multiple datasets (ATLAS-CC, ATLAS-Pes2o, Enron) with ablations (entity vs. event masking, pruning strategies, encoders) show the effectiveness of the method.

**Weaknesses:**

- KG2KV requires relation rewriting (e.g., "because" -> "cause"), which relies on external LLMs. This introduces dependency on external models and may limit reproducibility.

- The baselines are strong (KBLaM, RAG), but newer lightweight RAG variants (e.g., caching-based or hybrid symbolic-neural methods) are not included.

- The claim that only 20K KG2KV samples suffice for generalization is intriguing, but more discussion is needed on why such small-scale training suffices for billion-scale integration.

**Questions:**

- How sensitive is AtlasKV to the quality of relation rewriting in KG2KV? Could noisy or ambiguous relations degrade performance?

- How does AtlasKV handle evolving KGs (e.g., streaming updates)? Is retraining or re-encoding required?

- Could AtlasKV be combined with lightweight retrieval (hybrid parametric + non-parametric) for further efficiency gains?

---

> ### Author Response · Authors · 2025-11-19
> **Response to Reviewer bR9N (1/2)**
>
> We appreciate Reviewer bR9N for highlighting our work’s contribution and insightful feedback!
>
> >W1 & Q1. How sensitive is AtlasKV to the quality of relation rewriting in KG2KV.
>
> Thank you for your valuable feedback! We would like to first clarify that compared with the fully synthetic method in KBLaM, the relation rewriting task in KG2KV is a much easier task for LLMs. Because the Synthetic method needs human-predefined schemas as a longer context to generate all triples. However, the only context needed in KG2KV method is the original relation and the masked position. So the reproductivity of KG2KV is better than that of Synthetic.
>
> We also conducted experiments to analyze the sensitivity of AtlasKV to the quality of relation rewriting in KG2KV from two perspectives: (1) training data, (2) testing data.
>
> For training data, we trained the model on a naive version of ATLAS-Wiki-QKV (denoted as $\text{AtlasKV-Ntrain}$), in which we remove the relation rewriting process and directly use the combination of original relations and unmasked entities as the keys. Then we tested that model on ATLAS-Pes2o-QKV.
>
> |Method|$10^1$ Triples||$10^2$ Triples||$10^3$ Triples||$10^4$ Triples||
> |-|-|-|-|-|-|-|-|-|
> ||ACC@1|ACC@5|ACC@1|ACC@5|ACC@1|ACC@5|ACC@1|ACC@5|
> |KBLaM|50.0|$\underline{80.0}$|25.5|52.7|3.6|14.5|0.0|5.5|
> |AtlasKV w/o HiKVP|$\textbf{100.0}$|$\textbf{100.0}$|$\textbf{92.7}$|$\textbf{100.0}$|$\textbf{72.7}$|$\textbf{90.9}$|$\textbf{47.3}$|$\textbf{67.2}$|
> |AtlasKV-Ntrain w/o HiKVP|80.0|$\textbf{100.0}$|80.8|$\underline{94.2}$|$\underline{55.8}$|$\underline{82.7}$|$\underline{38.5}$|$\underline{57.7}$|
> |AtlasKV|$\underline{90.0}$|$\textbf{100.0}$|$\underline{87.3}$|92.7|52.7|70.9|16.4|49.0|
> |AtlasKV-Ntrain|80.0|$\textbf{100.0}$|71.2|86.5|42.3|57.7|21.2|34.6|
>
> As shown in the above table, without relation rewriting in the training data, although there is a slight performance drop in $\text{AtlasKV-Ntrain w/o HiKVP}$ and $\text{AtlasKV-Ntrain}$, our model can still maintain a high knowledge grounding accuracy.
>
> For testing data, we apply the above naive KG2KV process to ATLAS-Pes2o-QKV, which is our testing data. We represent this version of AtlasKV as $\text{AtlasKV-Ntest}$.
>
> |Method|$10^1$ Triples||$10^2$ Triples||$10^3$ Triples||$10^4$ Triples||
> |-|-|-|-|-|-|-|-|-|
> ||ACC@1|ACC@5|ACC@1|ACC@5|ACC@1|ACC@5|ACC@1|ACC@5|
> |KBLaM|50.0|$\underline{80.0}$|25.5|52.7|3.6|14.5|0.0|5.5|
> |AtlasKV w/o HiKVP|$\textbf{100.0}$|$\textbf{100.0}$|$\textbf{92.7}$|$\textbf{100.0}$|$\textbf{72.7}$|$\textbf{90.9}$|$\textbf{47.3}$|$\textbf{67.2}$|
> |AtlasKV-Ntest w/o HiKVP|$\textbf{100.0}$|$\textbf{100.0}$|76.9|$\underline{98.0}$|$\underline{55.8}$|$\underline{78.8}$|$\underline{26.9}$|46.2|
> |AtlasKV|$\underline{90.0}$|$\textbf{100.0}$|$\underline{87.3}$|92.7|52.7|70.9|16.4|$\underline{49.0}$|
> |AtlasKV-Ntest|$\underline{90.0}$|$\textbf{100.0}$|86.5|90.4|46.2|61.5|23.1|36.5|
>
> As shown in the above results, without relation rewriting in testing data, $\text{AtlasKV-Ntest w/o HiKVP}$ and $\text{AtlasKV-Ntest}$ can still perform well. The performance drops are also acceptable.
>
> This is mainly because **although more accurate key strings with relation rewriting in KG2KV are beneficial, simple combination of relation and unmasked entity strings can also be very helpful for distinguishing them**.
>
> We have added the above experiments into Appendix B.2 of our revision.
>
> > W2. Newer lightweight RAG variants are not included.
>
> We further compare AtlasKV with cache-augmented generation (CAG)[1], which is a new caching-based knowledge augmentation paradigm that preloads the LLM with all documents in advance and precomputes the KV cache.
>
> |Method|Time Complexity|Memory Complexity|
> |-|-|-|
> |ICL|$O((M\cdot T+N)^2\cdot D)$|$O((M\cdot T+N)\cdot(M\cdot T+N+D))$|
> |RAG|$O(M+R\cdot T+(R\cdot T+N)^2\cdot D)$|$O((R\cdot T+N)\cdot(R\cdot T+N+D))$|
> |KBLaM|$O((M+N)\cdot N\cdot D)$|$O((M+N)\cdot(N+D))$|
> |CAG|$O((R\cdot T+N)^2\cdot D)$|$O((R \cdot T+N)\cdot (N+D))$|
> |AtlasKV|$O((C_{t}\cdot \sqrt[3]M+N)\cdot N\cdot D)$|$O((C_{m}\cdot \sqrt[3]M+ N)\cdot(N+D))$|
>
> As shown in the above table, we compare the complexity during the inference process of AtlasKV with CAG and other baselines. Compared to the increased cost with every additional retrieved knowledge in CAG, **AtlasKV trades the sub-linear attention-based retrieval cost ($O(C_{t}\sqrt[3]M)$,$O(C_{m}\sqrt[3]M)$) for the ability to perform efficient inference independent of the scale of the retrieved relevant knowledge**. Moreover, **the cost of precomputing external knowledge in AtlasKV is also much smaller than that in CAG**. Because instead of precomputing the KV cache, AtlasKV only needs to obtain the embeddings through a lightweight sentence encoder.
>
> We have added this comparison into Section 5.2 of our revision.
>
> [1] Chan, Brian J., et al. "Don't do rag: When cache-augmented generation is all you need for knowledge tasks." Companion Proceedings of the ACM on Web Conference 2025. 2025.

---

> ### Author Response · Authors · 2025-11-19
> **Response to Reviewer bR9N (2/2)**
>
> Follows up to Response to Reviewer bR9N (1/2).
>
> > W3. More discussion is needed on why such small-scale training suffices for billion-scale integration.
>
> Thank you for the insightful comment. **The target of our training process is not to memorize specific knowledge in the training data, but to learn a projection from the pre-trained sentence encoder space to the LLM’s semantic space**. This alignment training process is easier than memorizing knowledge through brute-force fitting.
>
> KG2KV further improves the quality of training data by producing highly-diverse Q-K-V data that cover a broad spectrum of entity types and relation patterns. As some empirical results shown in Appendix E, the training dynamics reveal that AtlasKV starts to generalize by retrieving relevant knowledge from the external KG triples, instead of brute force over-ﬁtting through neural parameters, from a speciﬁc training step. In other words, the model learns to retrieve and align, not to memorise, which is why small-scale training suffices for billion-scale integration.
>
> > Q2. How does AtlasKV handle evolving KGs (e.g., streaming updates)? Is retraining or re-encoding required?
>
> Thank you for your valuable feedback. AtlasKV does not require retraining when the KGs evolve. The trained projector (attention heads $\tilde{\mathbf{W}}_Q$, $\tilde{\mathbf{W}}_K$, and $\tilde{\mathbf{W}}_V$) have already learned a semantics-preserving mapping that can generalize beyond the triples in the training data, which has been verified by the out-of-distribution evaluation in Section 5. Re-encoding the Q-K-V data from new triples is necessary, but it is lightweight. We only need to feed the Q-K-V strings into the frozen lightweight sentence encoder once and store the encoded embeddings. And the cluster tree can also be maintained by adding the new key vectors into their most related cluster through cosine similarities.
>
> > Q3. Could AtlasKV be combined with lightweight retrieval (hybrid parametric + non-parametric) for further efficiency gains?
>
> Thank you for this insightful suggestion! Yes, AtlasKV’s attention-based retriever can indeed be seamlessly combined with a lightweight (e.g., BM-25 or other dense retrievers) first-stage selector to make it more efficient. In this way, the lightweight retriever quickly prunes the candidate space to a smaller one. Then AtlasKV attention heads perform fine-grained, context-aware grounding on this shrunken set. This cooperation could further reduce the inference latency in AtlasKV and improve the efficiency.
>
> Beyond the retrieval process, we can also inject the knowledge through a hybrid way. Like REFRAG[2], we can use reinforcement learning to train a policy to determine which kind of knowledge needs to be augmented using non-parametric methods (e.g., ICL) and which needs to be augmented using parametric methods (e.g., AtlasKV), thereby achieving high accuracy, low latency, low token cost, and low GPU cost at the same time.
>
> Thanks again for your constructive improvement suggestions! This provides a very insightful and promising direction for future work based on AtlasKV!
>
> [2] Lin X, Ghosh A, Low B K H, et al. Refrag: Rethinking rag based decoding[J]. arXiv preprint arXiv:2509.01092, 2025.

---

> ### Author Response · Authors · 2025-11-27
> **Your feedback to our rebuttal would be greatly appreciated**
>
> Dear Reviewer bR9N,
>
> Thank you again for the constructive comments you gave us in your review. As the rebuttal phase will end on Dec 3, we would greatly appreciate it if you could also take some time to check if our rebuttal has addressed your concerns, and please let us know if you would like us to provide any further clarification about the concerns you have.
>
> Best,
>
> Authors

---

### Author Response · Authors · 2025-11-23
**General Response**

Dear Reviewers and AC,

We appreciate very much your time and constructive suggestions on our work! We would like to provide a summary of the strengths of our work commonly raised by the reviewers, as well as some key points which clarify the common concerns of the reviewers.

Thank the reviewers again for your recognitions and highlighting our work’s contributions! The common strengths are listed below:

- The proposed methods (KG2KV, HiKVP) in AtlasKV are **elegant, well-motivated, novel, and practical, which bridge symbolic KG structure with LLM attention in a natural way**. (bR9N, yVFP, K64s)
- AtlasKV provides a credible path to **scaling parametric knowledge augmentation to the billion-fact scale with sub-linear time and memory complexity, which has the impressive ability to handle 1B triples with <20GB VRAM**. (bR9N, yVFP, K64s, yJsc)
- AtlasKV shifts from retrieval-heavy RAG to parametric KG injection, which **addresses a critical problem**: **Efficiently and scalably augmenting LLMs with vast, external factual knowledge, which is a fundamental goal in AI**. (yVFP, yJsc)
- Extensive experiments demonstrates that AtlasKV achieves **clear improvements in both knowledge grounding and generalization**, with substantial reductions in memory cost compared to KBLaM and RAG. (bR9N, yVFP, K64s, yJsc)

---

> ### Author Response · Authors · 2025-11-23
> **Follows up to the general response**
>
> Follows up to General Response.
>
> Some key clarifications and improvements to our work based on the reviewers' concerns are listed below: (Changes in the revision are marked in blue)
>
> - Influence of relation rewriting in KG2KV. (bR9N, yJsc)
>     - We further conducted experiments to analyze the sensitivity of AtlasKV to the quality of relation rewriting in KG2KV from training data and testing data perspectives, respectively. And **without relation rewriting process in training or testing data, which is the only process that needs LLMs in KG2KV, AtlasKV can still perform well**. The performance drops are also acceptable.
>     - [Changes in the paper]: We have added implementation details, results, and discussions of the above experiments in *Appendix B.2*. We also added the discussions of the trade-off between accuracy and token costs in KG2KV, as  revealed in *Appendix F.3*.
> - Influence of knowledge injection frequency. (K64s)
>     - We further set the knowledge injection frequency $K$ as 1, 3, 10 and then compared the knowledge grounding performance of them. And we found **too frequent or infrequent knowledge injection will both lead to suboptimal performance**. $K = 3$ is the best choice for AtlasKV.
>     - [Changes in the paper]: We have added visualization results, implementation details, and discussions of the above experiments in *Appendix B.3*.
> - Inter-level correlation in top-k selection is not clear. (K64s)
>     - We further compared the knowledge grounding performance at the inter-layer as well as root-layer with different top-k selections at a higher layer. The results suggest that **as we appropriately increase the upper-layer top-k values, the lower-layer accuracy will continuously increase in most cases**. If too many keys are selected at the upper-layer, the accuracy at the lower-layer might also drop a little bit.
>     - [Changes in the paper]: We have added visualization results, implementation details, and discussions of the above experiments in *Appendix B.4.2*.
> - Benefits of knowledge coverage is not clear. (yJsc)
>     - We further compared the beneﬁts from knowledge coverage and the drawbacks from the noise introduced by larger KGs. And **the GPTScore of the responses generated with the larger KG is much higher than that of the responses generated with the smaller KG, which demonstrates that scales up the KGs is valuable for knowledge augmented LLMs**.
>     - [Changes in the paper]: We have added implementation details, results, and discussions of the above experiments in *Appendix B.5*.
> - Lack discussions about the hardware latency during real-world training/inference. (K64s)
>     - **During the training process, we do not need to train on such a large scale of KGs with HiKVP as described in Appendix A.1**. We further conducted experiments and the results demonstrate that **the latency during inference time will be sightly higher as the KGs become larger**.
>     - [Changes in the paper]: We have added implementation details, results, and discussions of the above experiments in *Appendix F.2*. We also included that into the limitations in *Appendix J*.
> - Concerns about the precision drop in HiKVP. (yJsc)
>     - Although there will be some performance drop in HiKVP for the AtlasKV's scalability, **as verified in Appendix B.2, the performance drop can be relieved by increasing the $k_{R}, k_{I}$ and $k_{L}$**.
>     - [Changes in the paper]: We have added discussions of the trade-off between accuracy and scalability of HiKVP in *Appendix F.1*.
> - Other changes in the revision.
>     - The training objective of AtlasKV has been further illustrated in *Section 4.2*. (K64s)
>     - We further compared the complexity of AtlasKV with that of cache-augmented generation (CAG) in *Section 5.2*. (bR9N)
>
> We remain open to further discussions to resolve any remaining concerns or address additional questions. Thank you for reviewing our paper again!
>
> Sincerely,
>
> Authors.

---

### Meta-Review · Area_Chair_SmtW · 2026-01-01

**Summary:**

This paper addresses the scalability challenge of augmenting large language models (LLMs) with large-scale knowledge graphs (KGs). It proposes AtlasKV, a scalable framework that (1) integrates a large KG  parametrically into selected attention layers and (2) applies hierarchical pruning to efficiently retrieve relevant KG triples at inference time. Empirically, AtlasKV demonstrates superior knowledge grounding accuracy and inference efficiency compared to representative baselines when scaling to billion-triple KGs.

**Reviewer Concerns:**

The main concerns raised by the reviewers focus on design details, such as relation rewriting and hyperparameter choices (K64s, bR9N, yJsc) and aspects of the experimental setup (yVFP, yJsc). These concerns are sufficiently  addressed through additional experiments, ablation studies, and clarifications provided by the authors during the rebuttal.

The remaining concerns primarily relate to inferior performance relative to baseline methods on small-scale KGs (yVFP) and accuracy degradation introduced by hierarchical pruning (yJsc). These limitations are clearly articulated by the authors and appropriately scoped. Importantly, they do not undermine the paper’s core contribution to scalable knowledge augmentation for LLMs in large-scale KG settings.

**Reviewer Scores:**

The two initially negative reviews (yJsc and K64s, both scoring 4) are likely to move to positive scores (6) following the rebuttal and discussion. Reviewer yVFP, who initially assigned a high score (8), raised conditional concerns during discussion, which were addressed in the authors’ follow-up. Taken together, the overall evaluation places the paper above the acceptance threshold.

---

### Decision · Program_Chairs · 2026-01-26

Accept (Poster)